# Cyclical dermal micro-niche switching governs the morphological infradian rhythm of mouse zigzag hair

Makoto Takeo [1], Koh-ei Toyoshima[1,2], Riho Fujimoto[3], Tomoyo Iga[1], Miki Takase[1], Miho Ogawa[2] & Takashi Tsuji [1,2] ✉

Biological rhythms are involved in almost all types of biological processes, not only physiological processes but also morphogenesis. Currently, how periodic morphological patterns of tissues/organs in multicellular organisms form is not fully understood. Here, using mouse zigzag hair, which has 3 bends, we found that a change in the combination of hair progenitors and their micro-niche and subsequent bend formation occur every three days. Chimeric loss-of-function and gain-of-function of *Ptn* and *Aff3*, which are upregulated immediately before bend formation, resulted in defects in the downward movement of the micro-niche and the rhythm of bend formation in an in vivo hair reconstitution assay. Our study demonstrates the periodic change in the combination between progenitors and micro-niche, which is vital for the unique infradian rhythm.

Biological rhythms play an important role in nearly all biological phenomena, from the beginning of development to the end of lifespan (reviewed in[1]). In vertebrate development, regionalization of the organ-forming field occurs concomitantly with somitogenesis, which is under the control of the segmentation clock ([2], reviewed in[3]). In the morphogenesis of periodic tissue/organ patterns, such as branching of the kidney ureteric tree and digital bone, which are vital for organs to function properly and efficiently, the oscillation of subsets of genes and signaling pathways responds to pattern formation ([4], reviewed in[5]). After birth, the maintenance of homeostasis and general activity are closely related to various biological rhythms, such as the sleep cycle and menstrual cycle, and the disruption of these biological rhythms causes diseases, including sleep disorders and mood disorders (reviewed in[1,6]). Moreover, with aging, the biological rhythm is attenuated and often exhibits a shift in phase, resulting in changes in physiological processes, such as behavior and temperature regulation (reviewed in[7]). Therefore, various biological rhythms have essential roles in development throughout life. However, past studies have mainly focused on physiological processes, and how periodic morphological patterns are maintained after birth is largely unknown.

In mammals, hair follicles (HFs) arise from embryonic hair germ and repeatedly regenerate throughout the lifespan in a process called the hair cycle. The HF is divided into two major regions: a permanent upper portion, including the bulge region, which is a niche for hair follicle epithelial stem cells (HFESCs), and a variable lower portion containing the hair bulb region, which is composed of hair matrix cells and dermal papilla cells (DPCs). Following hair follicle morphogenesis, the lower part of the hair follicle regresses (catagen phase), forms a hair germ-like secondary hair germ, and then becomes quiescent (telogen phase)[8]. During the regeneration phase (anagen phase), the lower part of the hair follicle, including the bulb region, is regenerated by the reciprocal interaction between HFESCs and DPCs containing mesenchymal stem cells. Hair matrix cells, which are progenies of HFESCs, differentiate into several types of cells and form a seven-layered hair follicle structure, which includes the outer root sheath, inner root sheath (IRS), and hair shaft. Recently, Yang and his colleagues demonstrated that DPc form four clusters of 'a micro-niches' and control the fate and behavior of hair matrix cells, contributing to the generation of a spatiotemporally coordinated heterogeneous lineage that gives rise to the complex structure of the follicles[9]. However, the

[1]Laboratory for Organ Regeneration, RIKEN Center for Developmental Biology (CDB) and RIKEN Center for Biosystems Dynamics Research (BDR), Hyogo 650-0047, Japan. [2]OrganTech Inc., Tokyo 104-0028, Japan. [3]Department of Bioscience, Graduate School of Science and Technology, Kwansei-Gakuin University, Hyogo 669-1337, Japan. ✉e-mail: takashi.tsuji@riken.jp

mechanism of hair regeneration, especially how the morphological pattern of the hair shaft is generated, is not fully understood.

The pelage on the mouse dorsal skin is composed of four types of hair shafts (guard, awl, auchene, and zigzag, which represent 1-3%, 30%, 0.1%, and 65-70% of the total number of hairs, respectively) that differ in length, thickness, number of medulla columns, and the presence and number of bends[10,11]. Zigzag hairs are the thinnest hair types among these types of hairs and consistently have 3 bends that turn in opposite directions, producing a wavy hair similar to that in some humans. Therefore, zigzag hair has attracted great interest as a suitable model for investigating the regulatory mechanism underlying periodical morphological patterns. Past studies have demonstrated the importance of the asymmetrical expression of *Shh* and its downstream target *Igfbp5* in the hair matrix across the dermal papilla during bend formation[12]. Ectopic expression of *Igfbp5* results in the thinning of hair shafts, while the suppression of *Igfbp5*-mediated effects by IGF signaling or overexpression of *Shh* results in straighter hair shafts[12,13]. These results clearly suggest the importance of these genes and pathways in the generation of hair bends in zigzag hair, yet the mechanism generating the rhythm of bend formation is largely unknown.

In this study, we identified that switching of the combination of the cell cycle-arrested hair matrix (G1-HM) cluster and dermal papilla cluster is vital for the periodic morphological changes in mouse zigzag hair. Time course analysis showed that bend points are generated every 3 days beginning at 9 days after anagen induction. During bend formation, changes in the arrangement of dermal papilla cells and the formation of G1-HM were observed once a day. In clear contrast, detailed spatiotemporal analysis of cell arrangement revealed that the cluster of DP micro-niche facing the G1-HM cluster switched every 3 days, a unique cycle of biological rhythm. Spatiotemporal gene expression analysis revealed that *Ptn* and *Aff3* are upregulated in G1-HM cells and the dermal papilla, respectively, on the day of bend formation. The deletion and over expression of either *Ptn* from keratinocytes or *Aff3* from dermal cells disrupted the timing of G1-HM cluster formation as well as micro-niche switching and subsequent bend formation. Collectively, our results indicate that cyclical niche switching, in addition to G1-HM cluster formation, is one of the key driving forces generating a unique rhythm in the periodic bend formation of mouse zigzag hair and show that cell dynamics need to be taken into account to understand the principles of progenitor/stem cell regulation by niche, periodic pattern morphogenesis, and unique biological rhythm generation.

## Results

### Bend formation occurs once every 3 days

Zigzag hairs have one column of medulla cells, a ladder structure containing the hair pigment, and three bends in the hair shaft (Fig. 1a, b). Based on these characteristics, zigzag hair is distinguishable from other types of hair shafts that have no or one bend and several columns of medulla cells (Fig. S1a and b). To determine how and when these bends are formed, we started by analyzing the morphological characteristics of the hair shafts of 7–8-week-old mice in detail.

Under the microscope, we observed thinning of the hair shaft and significantly wide spacing between each medulla cell at the bend (Fig. 1b, c in the middle panels) compared to that at other locations on the hair shaft (Fig. 1b left and middle panels and c). The cross section of the hair shaft showed an oval shape, and the direction of the long axis was rotated ~90° at the bend (Fig. 1b right panels, Supplementary Movie 1). The distance between each bend in the same hair shaft is ~1.5–1.8 mm, suggesting that bend formation occurs at a consistent time interval (Fig. 1d). Three-dimensional analysis of zigzag hair using confocal microscopy revealed that the hair shaft can be classified into two types showing a mirror image relationship, with the hair shaft between the 2nd and 3rd bend points as the central axis, in a ratio of ~1:3

(Fig. 1e, f, Supplementary Movie 2). We also found that the angle of each bend is ~160° (Fig. 1g). In 86-week-old mice, the distances between medulla cells and the bend angle both increased, which led to substantial variability in the 3D morphology of the zigzag hair, suggesting that the rhythm of bend formation timing becomes disrupted with aging (Fig. S2a–e).

To determine the timing of bend formation, we analyzed hair shafts beneath the skin. We found that thinning of the hair shaft and sparse distribution of medulla cells already occurred slightly above the bulb region, where the hair shaft is generated (Fig. 1h and i). To estimate the timing of bend formation, we synchronously induced hair shaft formation in all hair follicles within the mouse dorsal skin by depilation of the hair shaft and then measured the distance between the center of each bend and the bulb throughout the period of hair shaft formation (Fig. 1j). We found that the distance increased at a constant rate until 17 days post-hair depilation (dpd) and then reached a plateau (Fig. 1k). Based on the approximate line of the average value at each time point, we calculated the time when the distance between each bend and the hair bulb became zero, i.e., the time of bend formation, and estimated that the 1st, 2nd, and 3rd bends were formed at 9.0 days, 12.3 days, and 15.8 days after the induction of hair shaft formation, respectively. These results indicate that bend formation occurs according to an infradian rhythm with a 3-day cycle, which has not been reported for any other biological process.

### Cell dynamics change in the bulb prior to bend formation

To investigate the cellular mechanism underlying bend formation, because zigzag hair can be distinguished from other hairs by the presence of the 1st bend, we examined the cell dynamics of the hair bulb region at the 2nd bend formation using the H2B-EGFP/Fucci596 cell cycle reporter mouse model in which all nuclei are labeled by green fluorescent protein (GFP) and cells in $G_0/G_1$ phase are labeled by monomeric Kusabira-Orange2 (mKO2) fluorescent protein[14–16]. Hereafter, throughout the majority of the paper, we will refer to pseudo-time values based on the length of the hair shaft because the progression of the hair cycle and the timing of bend formation slightly vary among mice and among experiments, as seen in Fig. 1k.

A previous study identified that the dermal papilla (DP) is divided into four clusters of micro-niches (C1 to C4) and that each cluster acts as micro-niche, which determines the fate of adjacent hair matrix cells[9]. Hair matrix cells contacting micro-niche C2 or C3 give rise to the Gata3-expressing cuticle of the inner root sheath and Huxley's layer or the AE13-positive cuticle of the hair shaft and hair cortex, respectively (Fig. 2a). Three-dimensional analysis of individual hair follicles identified that DP micro-niche align as a monolayer at pseudo days 11.5 and 12.5 after the induction of hair depilation (P11.5 and P12.5 dpd), while they became multilayered at 12.0 dpd, 12 h before bend formation (Fig. 2b). To confirm this point, we analyzed multiple hair follicles by taking confocal images from the dorsal side of the skin samples and found that the percentage of multilayered DP micro-niche significantly increased at P12.0 dpd (Fig. 2c, d, Fig. S3a). Live imaging at 12 dpd revealed that the cell dynamics, instead of the cell division of the dermal papilla, are the driving force behind the formation of the multilayered DP micro-niche (Supplementary Movie 3). Interestingly, a subset of the hair matrix entered the $G_0/G_1$ phase, as evidenced by the expression of mKO2, and aggregated as the micro-niche became multilayered (Fig. 2e, f, Supplementary Movie 3). Although hair matrix cells became positive for mKO2 at the outer layer of the hair follicle as they differentiated, the number of hair matrix cells at $G_0/G_1$ phase adjacent to the micro-niche at P11.5 dpd was <6 cells, while it was increased by >6 cells in most hair follicles at P12.0 dpd (Fig. 2g). Therefore, in this study, we defined the timing of hair matrix cluster at $G_0/G_1$ phase (G1-HM) formation when the number of hair matrix $G_0/G_1$ phase facing the micro-niche was 6 cells or more. In hair follicles other than zigzag hair follicles, the population of G1-HM was also observed

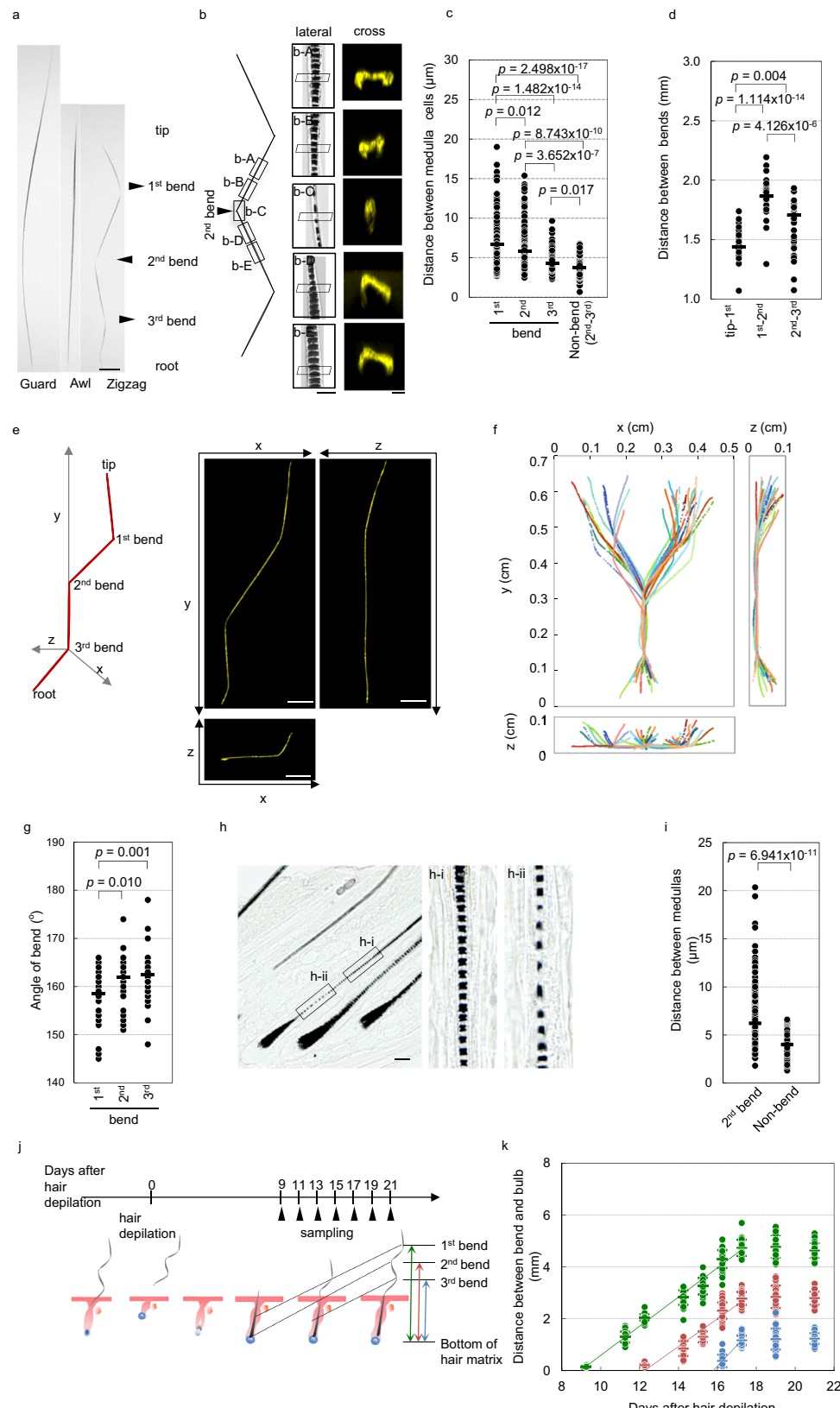

asymmetrically (Fig. S3b). However, quantification of awl hair folli-cles, which correspond to ~30% of total hair follicles[11], revealed that the number of hair matrix at $G_0/G_1$ phase facing the micro-niche was <6 cells, suggesting that G1-HM formation occurred only in zigzag hair (Fig. S3c). Three-dimensional analysis of the hair bulb region clearly showed that the G1-HM cluster is always formed

asymmetrically on one side facing the flat plane of the dermal papilla, which almost coincides with the left-right body axis (Fig. S3d and e). Immunohistochemical analysis for differentiation markers showed that G1-HM cells were positive for Gata3 but not AE13, suggesting that the HM that contacted micro-niche C2 became G1-HM (Fig. 2a and h).

**Fig. 1 | Bend formation occurs every 3 days by thinning and twisting of the hair shaft. a** Hair shaft shape of wild-type mice. **b** Medullar structure (lateral) and hair shaft shape in cross section (cross) of a zigzag hair at the indicated position. See also Supplementary Movie 1 and Fig. S1. **c** Quantification of the distance between each medulla cell of the zigzag hair (*n* = 25 hair shafts from 3 independent mice). **d** Quantification of the distance of each bend (*n* = 30 hair shafts from 3 independent mice). **e** Three-dimensional analysis of a zigzag hair shaft via confocal microscopy. See also Supplementary Movie 2. **f** Two-dimensional plot of a hair shaft generated from the 3D data shown in (**e**) (*n* = 30 hair shafts from 3 independent mice). **g** Quantification of the angle of each bend (*n* = 38 hair shafts from 3 independent mice). **h** Histological analysis of a hair shaft under the skin. The middle (h-i) and

right panels (h-ii) are high magnifications of the boxed area of the left panel. **i** Quantification of the distance between each medulla cell before eruption (*n* = 26 hair shafts from 5 independent mice). **j** Schematic of the time-course analysis of the distance between the bottom of the hair matrix and each bend. **k** Quantification of the temporal change in the distance between the bottom of the hair matrix and each bend (*n* = 475 hair shafts from 3 independent mice at each time point). The arrowheads in (**a**) indicate each bend. Quantification data show each data point (circle) and average value (horizontal bar). \*\**p* < 0.02, \*\*\**p* < 0.01 (two-tailed Student's *t*-test). Scale bars in (**a** and **e**), (**h**), (**b**, lateral), and (**b**, cross) indicate 500 μm, 50 μm, 20 μm, and 5 μm, respectively. Source data are provided as a Source Data file.

At 12.5 dpd, the IRS composed of Gata3- and mKO2-double-positive cells on only one side of the hair follicle became thicker, resulting in compression of the hair shaft and affecting the spatial distribution of the medulla cells (Fig. 2i, j. Supplementary Movie 4). Over time, the mKO2-positive cell population moved upward as the hair grew, and the compression of the hair shaft and altered distribution of medulla cells became more obvious at P13.0 dpd and kept moving upward (Fig. 2k, l). These results clearly suggest that cell dynamics in the DP and HM change before bend formation and that G1-HM is involved in bend formation (Fig. 2m).

### Micro-niche switching occurs only before bend formation
We next asked whether the change in cell dynamics of DP and HM occurs only before bend formation. To examine this point, we analyzed the bulb region from P10.0 dpd to P12.0 dpd, that is, from the period immediately following the 1st bend formation to the period immediately before the 2nd bend formation. Confocal microscopy revealed that the appearance of a G1-HM cluster composed of >6 cells occurred at P10.0 dpd and P11.0 dpd in addition to P12.0 dpd (Fig. 3a, b). To analyze the cell dynamics in greater detail, we generated surface plot data using high-resolution confocal microscopic images. In this analysis, we identified clear segregation of DP cells into four clusters, consistent with a previous report (Fig. 3a). We also found that the position of each cluster changed throughout the bend formation. From P11.0 dpd, the cells of each micro-niche become denser, the middle two clusters were close together, and the DP micro-niche appeared to be multilayered only at P12.0 dpd, consistent with Fig. 2b–d (Fig. 3a and c). Interestingly, we found that the G1-HM cluster faced micro-niches expressing Sema5a, a marker for micro-niche C3, as well as its lower cluster, micro-niche C2, at P12 dpd, while at P10.0 dpd and P11.0 dpd, the G1-HM cluster faced only C2; this pattern might be due to the positional change in C2 and C3 rather than G1-HM appearance at the new HM layer (Fig. 3a, d, and e)[9]. No obvious DP clusters or changes in the arrangement of G1-HM were observed between P11.0 dpd and P12.0 dpd in hair follicles other than zigzag hair follicles, such as awl hair follicles (Fig. S4). These results suggest that the change in the combination between G1-HM and DP micro-niches, indicating micro-niche switching, is the driving force of bend formation every 3 days (Fig. 3f).

### *Ptn* and *Aff3* are upregulated before bend formation
To identify the molecular mechanism underlying bend formation, we isolated >30 bulb regions by microdissection under a dissection microscope at 11.0, 11.5, and 12.0 dpd and compared the gene expression profiles at the different timepoints by RNA-Seq for two individual samples at each time point. Principal component analysis (PCA) confirmed that the overall gene expression profile differed among samples and that the samples were clearly distinguishable from each other by days after hair depilation (Fig. 4a). Differential expression analysis using EdgeR, a Bioconductor software package, revealed that 35 out of 321 differentially expressed genes (DEGs) were upregulated >2-fold at 12.0 dpd compared to 11.0 dpd, including *Shh*, which is known to control cell dynamics in multiple organs (Fig. 4b). To confirm the RNA-Seq data obtained using pooled bulb region samples, we

performed real-time qPCR on a bulb region isolated from a single hair follicle. We found that pleiotrophin (*Ptn*), a growth factor involved in the regulation of cell behavior and epithelial-mesenchymal interaction of multiple organs, and ALF transcription elongation factor 3 (*Aff3*), a transcription factor involved in multiple biological processes, were significantly upregulated only at 12.0 dpd, while *Shh* was upregulated at 11.0 and 12.0 dpd, when the G1-HG cluster was formed[17–22] (Fig. 4c).

Spatiotemporal gene expression analysis by whole mount in situ hybridization revealed that *Shh* is expressed in the G1-HM cluster at 11.0 and 12.0 dpd, consistent with the qPCR results (Fig. 4c, d). Because *Shh* is known as one of the circadian target genes and circadian genes have a role in hair production and oscillate in the hair matrix, *Shh* expression may be under the control of circadian rhythm[23–25]. In contrast to the gene expression pattern of *Shh*, the expression of *Ptn* is confined to the G1-HM cluster at 11.0 dpd and then expands broadly on both sides of the dermal papilla at 11.5 dpd and is restricted again to the G1-HM cluster at 12.0 dpd. The *Aff3* signal was found in DP micro-niche C3, consistent with a previous report, suggesting the important role of these genes in bend formation[9] (Fig. 4d).

### *Ptn* and *Aff3* are vital for the correct bending rhythm
To examine the functional significance of *Ptn* and *Aff3* in bend formation, we performed a hair reconstitution assay using the organ germ method in combination with gene modification by the CRISPR/Cas9 system[26,27]. We isolated skin epithelial cells and dermal fibroblasts, including hair follicle keratinocytes and dermal papilla cells, from E18.5 mouse embryos. We then removed the *Shh* gene as a control, the *Ptn* gene from the keratinocyte genome and the *Aff3* gene from dermal cells, followed by the generation of bioengineered hair germ (Fig. 5a). Since genes have the potential to affect hair follicle formation itself, we performed a chimeric analysis using a mixture of cells with and without incorporation of each guide RNA (Fig. 5b). By 14 days after transplantation of bioengineered hair germ, hair regeneration was observed in both the control and KO groups (Fig. 5c). The presence of CRISPR-modified gene alleles of *Shh*, *Ptn*, and *Aff3* was confirmed by Cyclave RCR and sequencing (Fig. S5a), suggesting that each gene was successfully knocked out at least in part. In the control groups for both epithelial KO and mesenchymal KO, 30.8 ± 15.0% and 16.5 ± 5.9% of zigzag hairs showed defects in bend formation, such as the smaller angle of bends, which might be due to artifacts (Fig. 5d, e). In clear contrast, the percentage of abnormal bend formation increased to 82.8 ± 7.6%, 82.5 ± 10.3% and 84.6 ± 5.4% in the *Shh*, *Ptn*, and *Aff3* KO groups, respectively (Fig. 5d, e). Quantitative analysis revealed that the variation in length between the hair tip and each bend was larger in the *Shh, Ptn* and *Aff3* KO groups than in the control group, resulting in a smaller number of bends and data points at the 2nd and 3rd bends in all KO groups in Fig. 5f (Fig. 5f, g). In the *Shh*, *Ptn*, and *Aff3* KO groups, 14.8%, 17.9% and 20.8% of zigzag hairs showed only 2 bends. Moreover, 7.7% of *Ptn* KO zigzag hairs had only one bend, suggesting that the timing of bend formation varies in the *Shh*, *Ptn* and *Aff3* KO groups, while no defect in the timing of bend formation was observed in either control group. (Fig. 5g). These results

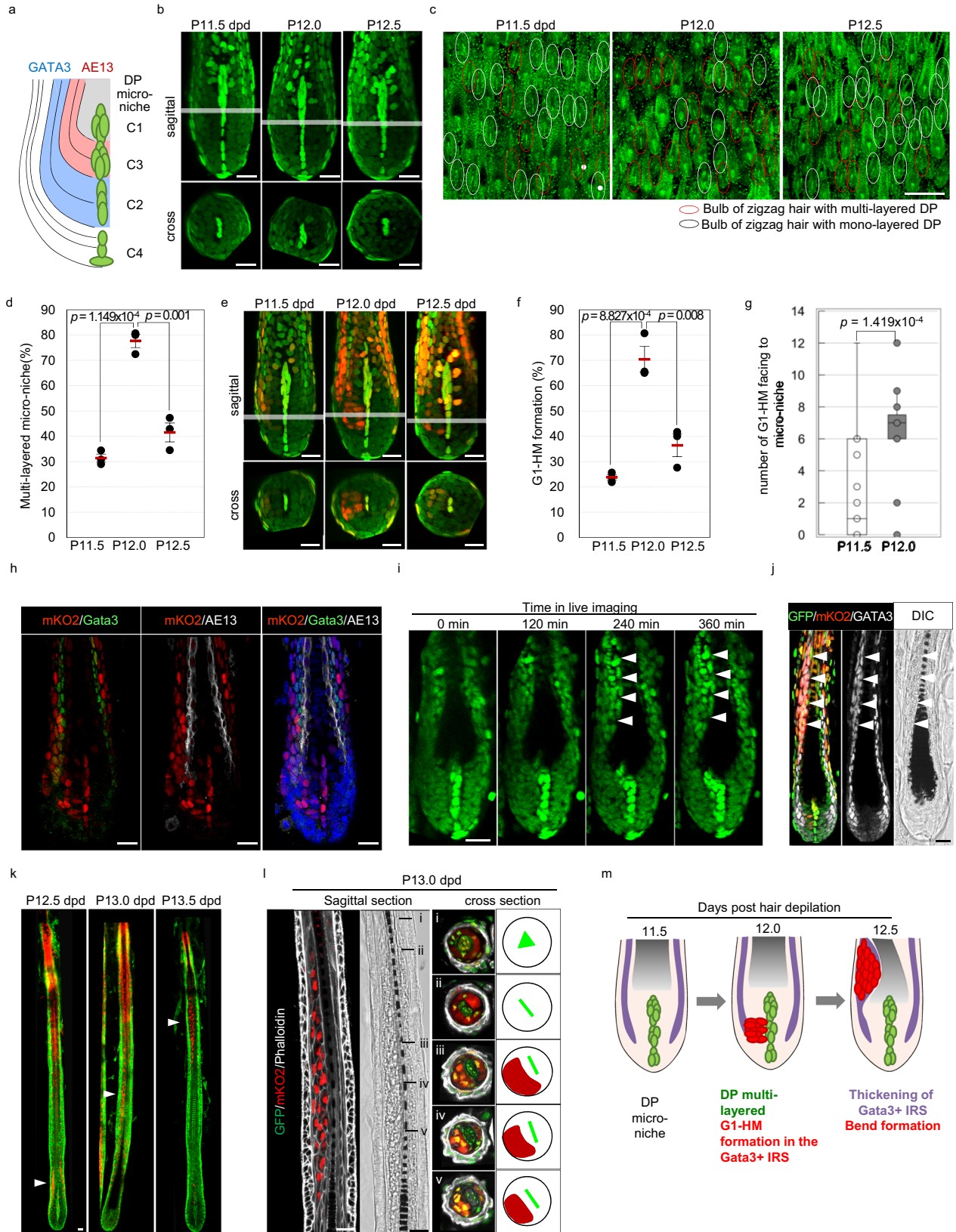

suggest that *Shh*, *Ptn* and *Aff3* have a role in determining the timing of bend formation potentially through affecting the rate of matrix cell proliferation between wildtype and KO cells because of chimeric KO assay.

Consistent with this observation, at P12.0 dpd, when the G1-HM cluster was observed in all control hair follicles, the G1-HM cluster was not observed in some hair follicles of the *Ptn* and *Aff3* KO groups (Fig. 5h). However, G1-HM cluster formation was observed other than on the P12 dpd in the *Ptn* and *Aff3* KO groups, when no G1-HM cluster appeared in the control groups (Fig. 5h). Moreover, at P12.0 dpd, the G1-HM cluster in the *Aff3* KO group only faced DP micro-niche C2, while all G1-HM clusters in the control, *Shh* KO, and *Ptn* KO groups

**Fig. 2 | Coordinated changes in the cell dynamics of the dermal papilla and hair matrix occur immediately before bend formation. a** Schematic illustration of the layer structure of the hair bulb, the clusters of the dermal papilla micro-niche (C1 to C4), and the expression pattern of marker genes (GATA3 and AE13). **b** Three-dimensional analysis of the DP cell arrangement of H2B-EGFP mice at the indicated pseudo time point (P). **c** Distribution of multilayered (red circle) and monolayered DP (white circle) at the indicated time points. **d** Quantification of the percentage of multilayered DP at the indicated time points (n = 109, 100, and 100 at P11.5 dpd, P12.0 dpd, and P12.5 dpd, respectively). **e** Analysis of cell kinetics using cell cycle reporter mice. See also Supplementary Movie 3. **f** The percentage of hair follicles with G1-HM cluster formation (n = 109, 100, and 100 at P11.5 dpd, P12.0 dpd, and P12.5 dpd, respectively). **g** Quantification of the number of G1-HM cells facing DP cells (n = 27 and 25 at P11.5 dpd and P12.0 dpd, respectively). Data are shown as box

and whisker plots. The bounds of the box plot indicate the 25th and 75th percentiles, the bar indicates medians, the whiskers indicate minima and maxima, dots indicate individual values. **h** Immunohistochemical analysis of the indicated markers at P12.0 dpd. **i** Live imaging of the bulb region at 12.5 dpd. Note the thickening of one side of the hair follicle. See also Supplementary Movie 4. **j** Immunohistochemical analysis of the indicated markers at P12.5 dpd. **k** Whole-mount analysis of zigzag hair follicles isolated from H2B-EGFP/Fucci596 reporter mice at the indicated timepoints. **l** Immunohistochemical analysis of the indicated markers and DIC images at P13.0 dpd. **m** Proposed model of cell dynamics in bend formation. Arrowheads indicate the asymmetric thickening of the hair follicle. Quantification data are shown as the mean ± s.d. in (**d** and **g**). ***p < 0.01 (two-tailed Student's t-test). Scale bars in (**c**) and (**b**, **e**, and **h**–**l**) indicate 500 μm and 20 μm, respectively. Source data are provided as a Source Data file.

faced both C2 and C3, suggesting a defect in the downward movement of C3 in the *Aff3* KO group (Fig. 5i).

To further investigate the role of *Shh*, *Aff3* and *Ptn* in bend formation, we overexpressed mTagBFP2-tagged Shh and Ptn fusion proteins in E18 embryonic keratinocytes and Aff3-mTagBFP2 fusion proteins in dermal fibroblasts using a lentivirus overexpression system and performed a hair reconstitution assay (Fig. 6a). By 14 days after transplantation, hair regeneration was observed from bioengineered hair follicle germ generated from the *Ptn*- and *Aff3*-overexpressing cells. In contrast, no hair regeneration occurred in the *Shh*-overexpressing cells, consistent with a previous report that *Shh* overexpression in a subset of interfollicular basal cells results in the suppression of embryonic hair follicle development[28]. Whole-mount analysis of regenerated hair follicles revealed the expression of mTagBFP2 protein in the hair bulb region including hair matrix and IRS or DPCs in the group with lentivirus infection into epithelial cells or dermal cells, suggesting the expression of *Ptn* and *Aff3* (Fig. 6b). In both control groups, virtually all regenerated hair shafts showed 3 bends (Fig. 6c and d). In clear contrast, >75% of hair shafts regenerated from the *Ptn*-overexpressing cells bent two times or less (Fig. 6c and d). These hair shafts were thin and showed a sparse distribution of medulla cells compared to that of control hair shafts, suggesting that defects in bend formation might be due to the uniform expression of *Ptn* genes in the bulb region and subsequent uniform thickening of IRS (Fig. 6e). In the Aff3 overexpression group, ~60% of regenerated hair shafts bent four times (Fig. 6c and d). These results clearly suggest that *Shh* and *Ptn* are required for bend formation, while *Aff3* plays a role in maintaining the proper rhythm of bend formation by controlling the rearrangement of the micro-niche.

## Discussion

Periodic morphological patterns have a crucial role in the proper function of organs; thus, intense investigation is underway to elucidate the mechanism underlying the rhythms that control repetitive tissue morphology. In this study, we showed that switching the combination of the G1-HM cluster, which has characteristics similar to those of the signaling center, and the DP micro-niche involving *Aff3* and *Ptn* in association with the formation of the G1-HM cluster via *Shh* is vital for periodic morphological changes in zigzag hair regeneration in adult mice (Fig. 6f). Our results demonstrates that the combination between progenitor cells and their niche changes cyclically, governing the generation of periodic morphological rhythms.

Biological rhythms are vital for almost all biological processes, including morphogenesis, such as branching of the kidney ureteric tree and digital bone, during embryogenesis and homeostasis after birth, and their disturbance is closely associated with various malformations, diseases, and aging. In pattern formation of digit bone, several genes associated with Notch, Wnt, and Fgf signaling oscillate in a 3-hr cycle in the specialized group of cells in the apical ectodermal ridge (AER) and zone of polarizing activity (ZPA) (reviewed in[5]). The mechanism underlying circadian regulation is being elucidated and

can be explained in most cases by the transcription-translation feedback loop (TTFL) model[29,30]. In clear contrast, the mechanisms underlying morphogenesis showing infradian rhythms, as seen in deer antler, are poorly understood[31]. In the present study, we demonstrated that the morphological rhythm of mouse zigzag hair shafts is a stable rhythm with a period of 3 days established after birth, which represents a unique infradian rhythm. In the functional assay, bends in zigzag hair formed with a 3-day cycle according to the time after implantation of bioengineered hair follicle germ regardless of the hair cycle of the host animals (Fig. 5c–f). Moreover, in wild-type mice, zigzag hair bends three times, while auchene hair bends only one time and with different timing, even though the two hair types are located very close to one another, strongly suggesting that the local and hair type-specific oscillator generates the infradian rhythm of mouse zigzag hair[10]. In addition, the variability of the distance between each bend, i.e., the timing of bend formation, increased in aged mice compared to young mice, indicating the possibility of a close relationship between a disturbance of the morphological rhythm and aging, as seen in other biological processes as described above (Fig. S2). Similar to that of the mouse hair shaft, the morphological pattern of the human hair shaft is determined in the hair bulb, and curly hair is caused by the asymmetric thickening of the outer root sheath, the outermost layer of the hair follicle[32]. As seen in mouse zigzag hair, hair shafts in elderly people are less synchronized, which is part of the reason that hair waves become unaligned, and hair appears more frizzy, one of the most obvious characteristics of aging[33]. Taken together, our data will help elucidate the mechanism of generating the morphological infradian rhythm of the hair shaft and the relationship between its disturbance and aging not only in mice but also in humans.

Homeostasis of the organism after birth relies on the spatio-temporally well-coordinated behavior of stem/progenitor cells, which is controlled by intrinsic mechanisms as well as extrinsic mechanisms by their niche through direct cell–cell interactions and short-range molecular signals emitted from the niche[34–36]. In the adult stem cell-niche system, such as hematopoietic stem cells and stroma cells and Lgr5+ intestinal stem cells and Paneth cells, displacement from the niche by asymmetric division or cell migration triggers behavioral changes in stem/progenitor cells, resulting in the generation of various types of differentiated cells[34–37]. Hair follicles harbor multiple types of stem cells and their niches, such as sebaceous glands, arrector pili muscles, melanocytes and organ-inductive epithelial and mesenchymal stem cells located in the bulge and dermal papilla, respectively[30,38–41]. The coordinated behavior of these stem/progenitor cells is vital for controlling the spatiotemporal arrangement of each progeny and establishing the proper morphology of the hair follicle and hair shaft. Recently, in addition to cell displacement from the niche, Yang and his colleagues demonstrated the existence of sub-regions, referred to as 'micro-niches', in the hair bulb niche that contribute to the generation of a heterogeneous lineage composing the hair follicle and hair shaft[9]. Nevertheless, in any of these cases, the location of a certain niche within the tissue is physiologically well

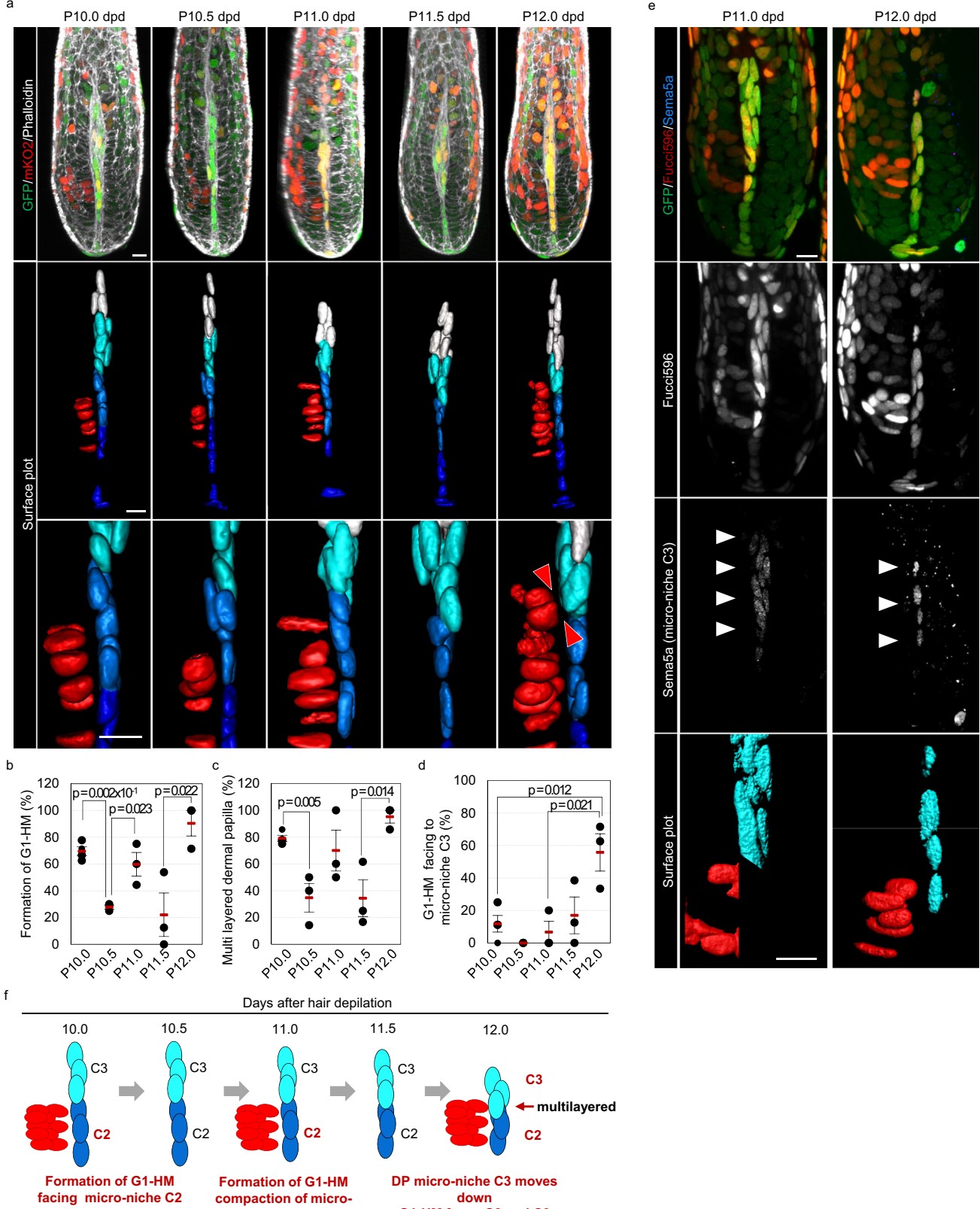

**Fig. 3 | The DP micro-niche facing the G1-HM cluster shifts only before bend formation. a** Immunohistochemical analysis of the indicated markers and 3D surface plot analysis over the entire time course of 2nd bend formation. C1–C4 indicate the cluster of DP micro-niche. Note that G1-HM faces micro-niche C3 only at P 12.0 dpd (arrowheads). **b**–**d** Quantification of the percentage of hair follicles showing G1-HM cluster formation (**b**), multiple layers of the uppermost part of micro-niche C2 (**c**), and G1-HM cluster facing micro-niche C3 (**d**) (n = 33, 25, 22, 27,

and 25 from 3 independent mice at P10.0, P10.5, P11.0, P11.5, and P12.0 dpd, respectively). **e** Immunohistochemistry against Sema5a, a marker for micro-niche C3, and surface plot of Sema5a-expressing cells (blue) and G1-HM (red). **f** Schematic summary of the time-course analysis of cell dynamics during entire bend formation shown in (**a**). Quantification data are represented as the mean ± s.d. *$p < 0.05$, **$p < 0.02$, ***$p < 0.01$ (two-tailed Student's t-test). Scale bars indicate 10 μm. Source data are provided as a Source Data file.

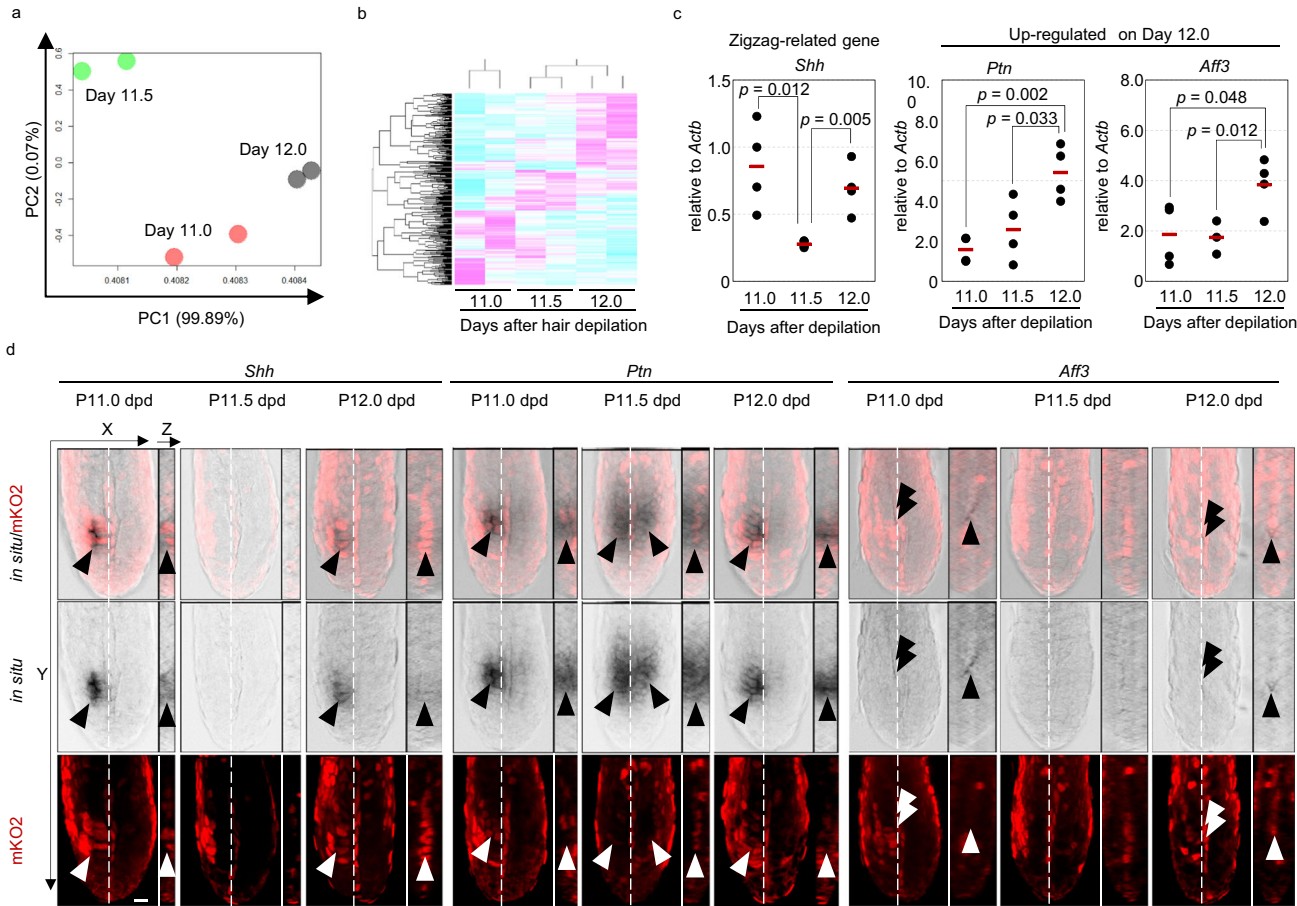

**Fig. 4 | *Ptn* and *Aff3* are upregulated in the bulb only before bend formation.**
**a** Principal component analysis of the gene expression profile of pooled whole bulb regions at 11.0, 11.5, and 12.0 dpd (*n* = 2 pooled bulbs from individual mice at each time point). The proportions of variance of PC1 and PC2 are 99.89% and 0.07%, respectively. **b** Clustering analysis of differentially expressed genes identified by the EdgeR Bioconductor software package. **c** qPCR of the indicated genes (*n* = 4

single bulbs at each time point). **d** Representative whole-mount in situ hybridization images for the indicated genes at the indicated time points. The dotted line indicates the position of the z slice. The arrowheads indicate the region of gene expression. Quantification data show each data point (circle) and average value (horizontal bar). *$p < 0.05$, **$p < 0.02$, ***$p < 0.01$ (two-tailed Student's *t*-test). Scale bars indicate 10 µm. Source data are provided as a Source Data file.

defined and stable, and the combination between certain stem/progenitor cells and their niche is unchangeable. In this study, we demonstrated that the arrangement of specialized hair matrix cells, e.g., G1-HM, and their DP micro-niches changes, and this change is tightly correlated with the generation of hair morphological patterns. We show that a DP micro-niche repetitively moves downward and upward according to the infradian rhythm over a 3-day period and changes the combination of micro-niche faces the G1-HM cluster, altering the differentiation status of the Gata3-positive IRS layer and leading to the formation of the bend of zigzag hairs (Figs. 2, 3 and 5). These results demonstrate that the location of micro-niches is not static but can change periodically (so-called 'niche switching') and trigger behavioral changes in stem/progenitor cells with the appropriate timing and in the correct locations. Our findings indicate that the location of the micro-niche is dynamic, illustrating the concept that periodic change in the combination of stem/progenitor cells and their niche contributes to the heterogeneity of their progenies and rhythmic patterning of tissue/organ morphology.

The behavior of stem/progenitor cell and tissue/organ morphogenesis is achieved by complex gene expression patterns and growth factors secreted from specialized cell populations, referred to as signaling centers, such as enamel knots in teeth and zones of polarizing activity (ZPA) in limb development[42,43]. *Shh* is one of the key factors expressed in the signaling center and is involved in the morphogenesis of multiple organs[42,43]. Previous studies of bend formation in zigzag

hair reported the importance of *Shh*, which is expressed asymmetrically in the bulb under the control of WNT and ectodysplasin (EDA) signaling and regulates the expression of downstream components such as *Igfbp5* and *EGR2* (*Krox20*)[12,13,44]. Symmetrical expression of *Shh* in the entire bulb in *Foxn1:Shh* transgenic mice results in a thin hair shaft with a smaller bend angle[12]. In the current study, *Shh* expression was confined to specialized hair matrix cells, G1-HM cells, which were in the $G_0/G_1$ stage of the cell cycle. Partial knockout of the *Shh* gene resulted in less thickening of one side of the IRS, and the bend showed a smaller curve angle. This might be because the IRS of the *Foxn1:Shh* mice was uniformly thickened, whereas less thickening occurred in the *Shh* KO mice, suggesting that *Shh* is responsible for the formation of the G1-HM cluster and the thickening of IRS. In clear contrast with *Shh*, *Ptn* and *Aff3* are upregulated at the time of bend formation in the G1-HM cluster and DP micro-niche C3. *Ptn* is associated with multiple biological processes, including cell proliferation and migration in multiple organs. In mammary gland development in mice, *PTN* is expressed in the mammary gland epithelium, and treatment with an anti-PTN antibody enhances the ductal development of the mammary gland, implying that *PTN* suppresses epithelial cell differentiation and promotes cell proliferation[19]. Similarly, in bend formation, *Ptn* may suppress the differentiation of G1-HM cells, enabling these cells to proliferate more, resulting in the thickening of the IRS, which is vital for bend formation. *Aff3* is also involved in multiple biological processes, including axis formation, and the injection of human *AFF3*

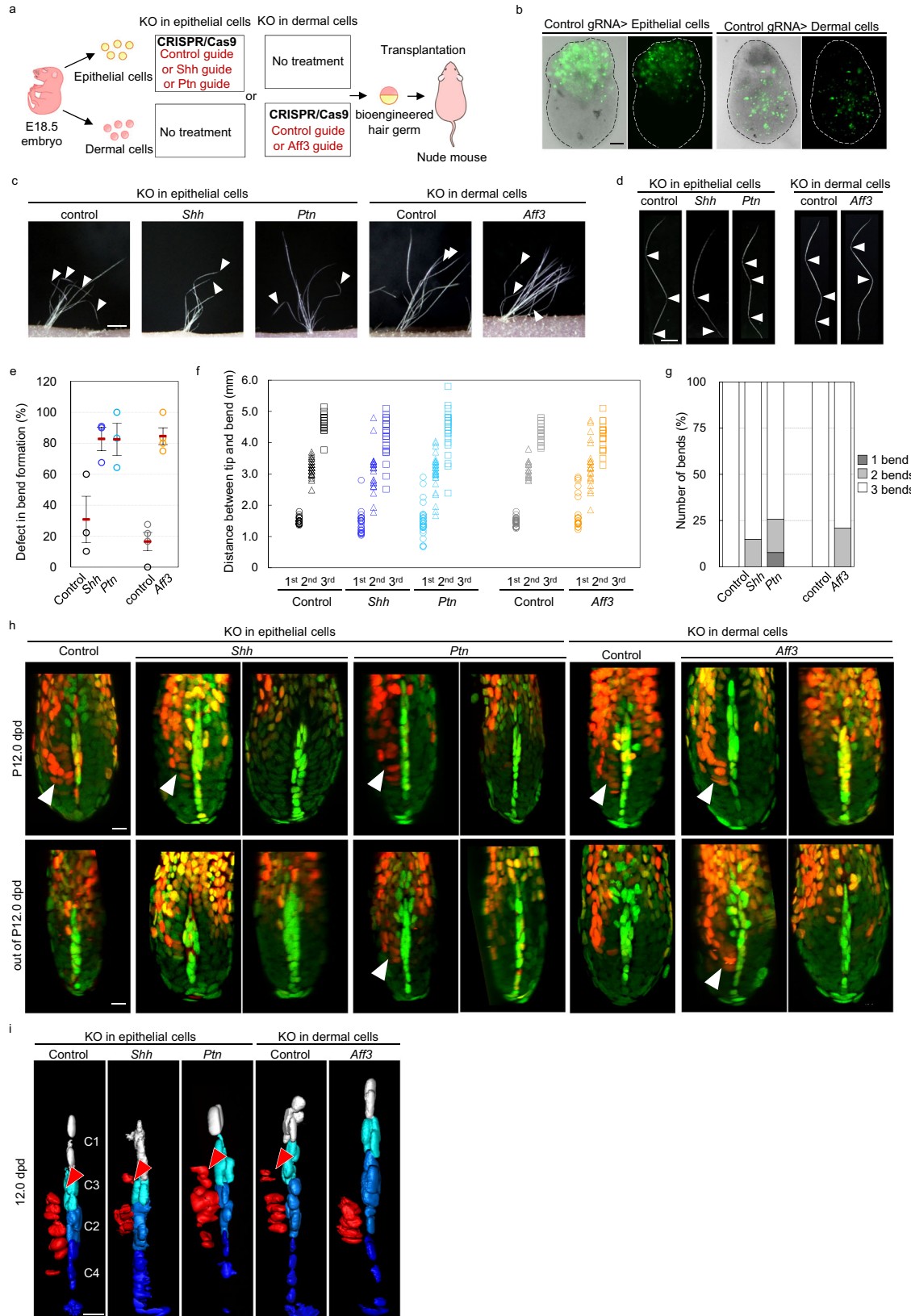

mRNA in zebrafish embryos results in severe defects in the body axis, suggesting a defect in the rhythm of somitogenesis[45]. In the current study, partial KO or overexpression of *Aff3* in dermal cells impaired niche switching and subsequent bend formation, although *Ptn* expression in G1-HM was intact, suggesting that a change in the DP micro-niche, which might be under the control of *Aff3*, affects the

behavior of *Ptn*-positive G1-HM and subsequent bend formation (Fig. 5). Although the detailed mechanism of how *Aff3* generates the infradian rhythm should be further investigated, our results indicate that the morphological rhythm of mouse zigzag hair is generated by changes in the inner hair root sheath through *Shh* and *Ptn* and not by changes in the hair shaft itself under timing control by the *Aff3* gene.

**Fig. 5 | Chimeric knockout of *Ptn* or *Aff3* disturbs the cyclical pattern of bend formation. a** Schematic of functional analysis by a combination of chimeric gene knockout via CRISPR/Cas9 and the organ germ method. **b** Representative image of bioengineered organ germ using epithelial cells or dermal cells introduced with GFP-labeled control guide RNA. **c** Hairs generated from the bioengineered organ germ using the indicated cells after transplantation in immune-deficient mice. **d** Morphological analysis of zigzag hair. **e**–**g** Quantification of the ratio of zigzag hair with defects in bend formation (**e**) ($n = 63, 58, 50, 83$, and 48 hair shafts for the control for epithelial KO, *Shh* KO, *Ptn* KO, the control for dermal KO, and *Aff3* KO, respectively), the distance between the hair tip and each bend (**f**) ($n = 29, 28, 39, 31$, and 24 hair shafts for the control for epithelial KO, *Shh* KO, *Ptn* KO, the control for dermal KO, and *Aff3* KO, respectively), and the ratio of hair follicles having the indicated number of bends (**g**) ($n = 29, 28, 39, 32$, and 25 hair shafts for the control for epithelial KO, *Shh* KO, *Ptn* KO, the control for dermal KO, and *Aff3* KO, respectively). **h** Whole-mount analysis of G1-HM cluster formation at P 12.0 dpd or other timepoints (out of P12.0 dpd). Note the G1-HM cluster formation in *Ptn* and *Aff3* chimeric KO hair follicles not only at P12.0 dpd but also at other timepoints. **i** Surface plot analysis of dermal papilla cells in hair follicles under the indicated conditions at P12.0 dpd. Note the defect in downward migration of micro-niche C3 in *Aff3* KO hair follicles. The arrowheads in (**c**), (**d**), (**g**), and (**i**) indicate the regenerated zigzag hair, bend point, G1-HM cluster formation, and G1-HM facing micro-niche C3. Quantification data are represented as the mean ± s.d. in (**e**) and each data point in (**g**). Scale bars in (**c**), (**d**), (**b**), and (**h** and **i**) indicate 1 mm, 500 μm, 100 μm, and 10 μm. Source data are provided as a Source Data file.

In conclusion, our study show that the rhythm of periodic morphological patterns in mammalian tissues/organs after birth is maintained by dynamic changes in the combination of progenitor cells and their micro-niches. The universality and importance of niche switching in generating infradian rhythms and in stem cell regulation must be further investigated. Our study illustrates a concept, 'niche switching', and will contribute to the fields of developmental biology, stem cell biology, and chronobiology.

## Methods

### Animals
The C57BL/6NCrSlc, BALB/cCrSlc, BALB/cSlc-*nu/nu* mice were purchased from SLC Inc., Japan. The R26-H2B-EGFP mouse line (CDB Acc. No. CDB0238K, https://large.riken.jp/distribution/reporter-mouse.html) and B6.Cg-Tg(FucciG1)#596Bsi mouse line (BRC No. RBRC02707, https://knowledge.brc.riken.jp/resource/animal/card?_lang_=en&brc_no=RBRC02707) were kindly gifted from Laboratory for Animal Resources and Genetic Engineering (LARGE), RIKEN Center for Biosystems Dynamics Research, and RIKEN BioResource Research Center (BRC), respectively[14–16]. The animals were housed in 12 h light/12 h dark condition, 21–25 °C and 40–60% humidity, and all the experimental procedures using animals were approved by the Institutional Animal Care and Use Committee of RIKEN Kobe Branch and performed in accordance with the relevant guidelines and regulations.

### The 3D analysis of mouse zigzag hair shafts
Zigzag hair shafts were harvested from the back skin of C57BL/6NcrSlc mice or R26-H2B-EGFP/ B6.Cg-Tg(FucciG1)#596Bsi double transgenic mice at age of 7–8 and 86 weeks old. Three-dimensional image capture was performed using a LSM 780 and LSM 880 confocal microscope (Carl Zeiss, Germany). Cross section image of hair shaft was generated from the 3D image using Imaris software Ver. 7.6.5 (Oxford Instruments, UK). The 2D plot was created by plotting the xy, xz, and yz coordinates at >257 points for each hair shaft on Microsoft Excel (Microsoft, USA).

### Live imaging analysis of whole hair bulb
Live imaging was performed using a LSM 880 confocal microscope with Airyscan (Carl Zeiss, Germany). Under isoflurane anesthesia, the skin on the body side of R26-H2B-EGFP/ B6.Cg-Tg(FucciG1)#596Bsi double transgenic mice was incised 2-3 cm in size along the hair flow, and imaging was performed from the exposed surface of skin section. Images were acquired with 0.57 or 0.58 μm step for the z-axis at 30-min intervals for 4 or 6 h.

### Immunohistochemistry
Fluorescent immunohistochemistry was performed on 4% PFA fixed-paraffin sections (10 μm-thickness), frozen sections (50 μm-thickness), and whole-mount hair follicles. Following deparaffinization and rehydration, paraffin sections were incubated in 1 mM EDTA for 40 min at 95 °C for antigen retrieval. No antigen retrieval was performed on frozen sections and whole-mount samples. Specimens were treated in blocking solution (PBS containing 1% w/v bovine serum albumin (BSA) and either 0.1 or 0.5% Triton-X 100 in PBS (137 mM NaCl, 8 mM Ha$_2$HPO$_4$, 2.7 mM KCl, 2 mM KH$_2$PO$_4$) for paraffin section or frozen section and whole-mount samples) for 1 h at room temperature, then incubate in primary antibodies diluted with blocking solution for overnight at 4 °C. After wash with 0.1% or 0.5% Triton-X 100 in PBS for 5 or 1 h for 3 times at room temperature, reactions with secondary antibodies and, if necessary, subsequent Avidin-Streptavidin amplification were accomplished. All fluorescence microscopy images were obtained with an LSM 780 and LSM 880 confocal microscope (Carl Zeiss, Germany). The antibodies used for immunohistochemistry are listed in Supplementary Data. 1.

### RNA-sequence and data analysis
Total RNA was extracted from hair bulb pool containing at least 30 bulbs isolated by microdissection using the ReliaPrep FFPE Total RNA MiniPrepSystem (Promega, USA) from two individual animals for each time point. SMART-Seq Strand RNA library preparation, RNA-Sequence, and general data processing was performed using NovaSeq RNA-seq analysis service (TAKARA BIO Inc., Japan). Data analysis was performed using R (version 4.0.5) and RStudio (version 1.4.1106) using the count matrix normalized by Transcripts per million method provided from TAKARA BIO Inc.. PCA was conducted by prcomp function of stats package and 3-dimensional plot were visualizes using plot3d function. Relative gene expression level was compared using EdgeR package (version 3.42.4) and genes showing FDR < 0.05 were considered as differentially expression genes. Relative gene expression levels were described as Z-score calculated using genescale function, then heatmap was constructed using heatmap.2 function of gplots package (version 3.1.3). Data set have been deposited in the Gene Expression Omnibus (GEO) database under accession codes GSE211948.

### qPCR
RNA extraction, cDNA synthesis, and first amplification was performed using CellAmp Whole Transcriptome Amplification Kit (Real Time) ver. 2 (TAKARA BIO INC., Japan). Real-time qPCR was performed on the Applied Biosystems QuantStudio 12 K Flex (Life Technologies, USA) using SYBR Premix Ex Taq II (TAKARA BIO INC.). The data were normalized to β-actin expression. The primer pairs used for real-time qPCR are listed in Supplementary Data. 1.

### Whole mount in situ hybridization
Whole mount in situ hybridization was performed based on the previously described method with several modifications[46]. Mouse back skin was cut into small pieces containing ~3 × 3 hair follicles and fixed in 4% PFA in PB (19 mM NaH$_2$PO$_4$, 81 mM Na$_2$HPO$_4$) at 4 °C for overnight, dehydrate, and store in 100% Methanol at −30 °C. After rehydration, adipocyte layer of skin samples was removed as much as possible using fine forceps and incubated in 6% hydrogen peroxide in PBT (0.1% Tweeen 20 in PBS) for 1 hr at 4 °C with rotation. Specimens were the treated with 10 μg/ml Proteinase K (Roche, Switzerland) for 10 min at

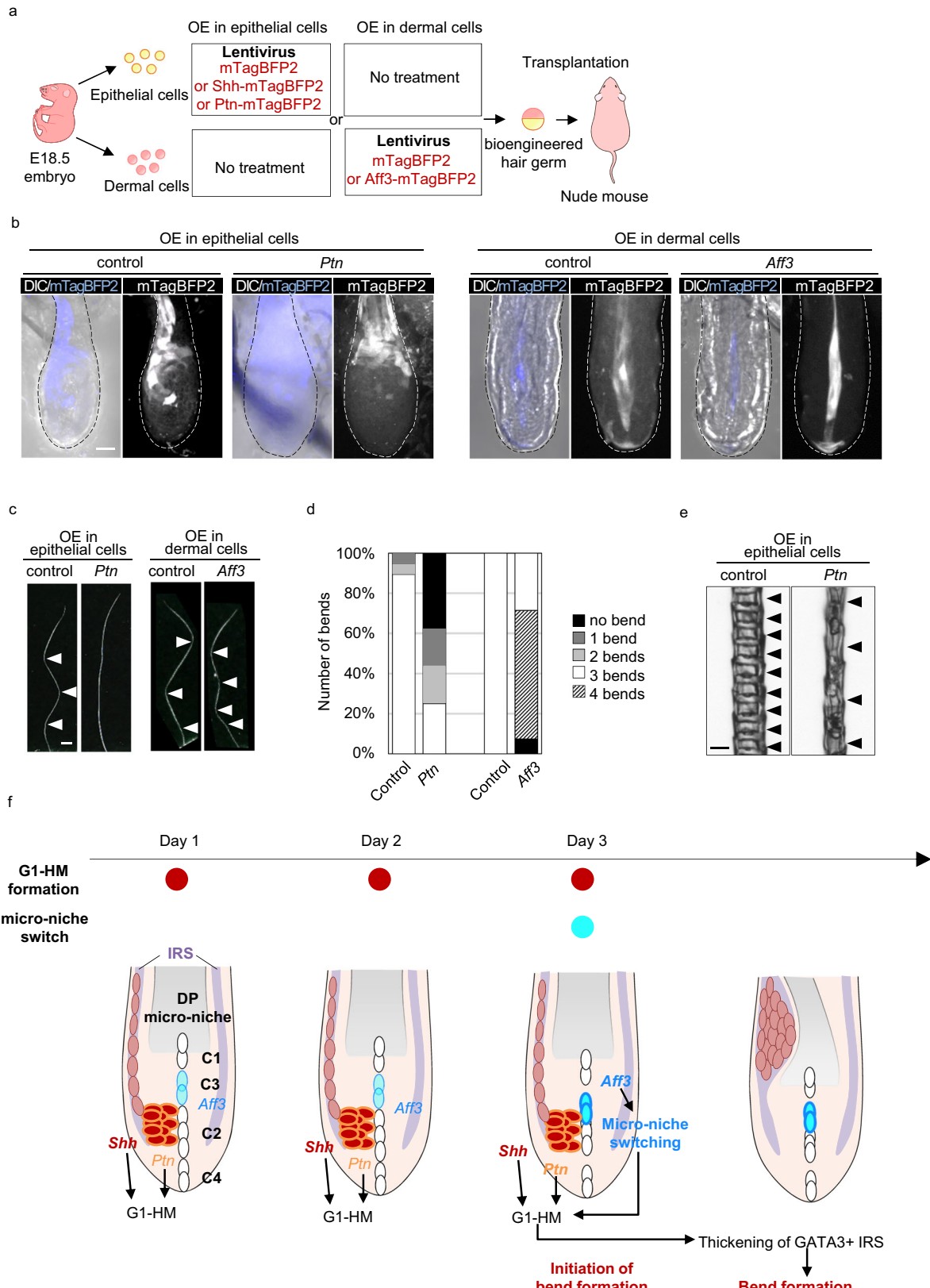

**Fig. 6 | Forced activation of the *Ptn* or *Aff3* gene causes defects in bend formation. a** Schematic of functional analysis by a combination of gene over-expression (OE) via the lentivirus system and the organ germ method. **b** Whole-mount analysis of the expression of the mTagBFP2-tagged fusion protein. **c** Morphological analysis of zigzag hair. **d** Quantification of the ratio of hair follicles having the indicated number of bends (*n* = 25, 31, 30, and 24 hair shafts for the control for epithelial OE, *Ptn* OE, the control for dermal OE, and *Aff3* OE, respectively). **e** Morphological analysis of regenerated hair shaft. **f** Proposed model of the cellular and molecular mechanism of bend formation in mouse zigzag hair. The dashed line in (**b**) indicates the outline of the hair bulb. Arrowheads in (**c**) and (**e**) indicate the bend point and medulla structure, respectively. Scale bars in (**b**), (**c**), and (**e**) indicate 20 µm, 200 µm, and 10 µm. Source data are provided as a Source Data file.

37 °C, re-fix with 0.2% glutaraldehyde and 0.1% Tween 20 in 4% PFA/PB for 20 min at 4 °C. After prehybridization, hybridization was performed in hybridization solution (50% Formamide, 5X SSC, 1X Denhardot's solution, 10 mM EDTA, 0.1% Tween20, 50 μg/ml tRNA) containing 250–1000 ng/ml digoxigenin-labeled RNA probe at 50 °C for overnight. Following several wash steps, specimens were incubated in alkaline phosphatase conjugated anti-digoxigenin antibody at 4 °C for overnight. Signals were developed in NTMT (100 mM NaCl, 100 mM Tris–HCl (pH 9.5), 50 mM $MgCl_2$, and 0.1% Tween 20 final 2 mM Levamisole in 12% poly(vinyl) alcohol) solution containing NBT/BCIP (Roche, Switzerland) solution. After color development, whole-mount immunohistochemistry for Kusabira-Orange2 (mKO2) was performed.

## Functional analysis

Skin epithelial cells and dermal cells were isolated from E18.5 mouse embryo according to the method reported previously[27]. In brief, embryonic backs kin were treated with 4.8 U/ml dispase (BD Biosciences, USA) in PBS at 4 °C for 1 h, and then separated into epithelial and mesenchymal layers. Epithelial layer was incubated in Accutase (gibco, USA) at room temperature for 45 min and filtrated through 35 μm cell strainer to obtain single cell suspension. The dermal layer was digested in 10,000 U/ml collagenase (Worthington, USA) for 1 h at 37 °C with shaking and filtrated through 10 μm cell strainer. Gene knockout for both types of cells was achieved by CRISPR/Cas9 system using TransIT-Pro Transfection kit (TAKARA BIO INC., Japan) and guide RNAs that were designed using CHOPCHOP v3[47], as listed in Table S1, according to the manufacture's instruction. Epithelial cells and dermal cells were reconstituted into bioengineered hair follicle germ using the organ germ method and transplanted into BALB/cSlc-*nu/nu* mice[26]. Regenerated hair shafts and hair follicles were analyzed as described above.

## Detection of CRISPR-modified gene allele

Genomic DNA was extracted from the hair follicle by boiling in 50 mM HaOH at 98 C for 10 min. Using this genomic DNA as template, the sequence including gRNA region was amplified by PCR and cloned into pBlueScript II SK (+) plasmid. DNA sequence was determined by using cloned samples.

## Generation of lentivirus expressing mTagBFP2-tagged Shh, Ptn, or Aff3 gene

Open reading frame of Shh, Ptn, and Aff3 were amplified from mouse cDNA library and cloned into pLVSIN-CMV-neo Vector with GFP and IRES sequence. Virus solution was produced by co-transfection of lentivirus vector and Lentiviral High Titer Packaging Mix into Lenti-X 293 T cells using TransIT-293 Transfection Reagent (TAKARA BIO INC., Japan).

## Statistics and reproducibility

Student's *t*-test was used to calculate *P*-values on Microsoft Excel, with two-tailed tests. Each experiment was repeated at least three times independently and obtained similar result.

## Reporting summary

Further information on research design is available in the Nature Portfolio Reporting Summary linked to this article.

## Data availability

All graph data generated in this study are provided in the Source Data file. RNA-seq data set have been deposited in the Gene Expression Omnibus (GEO) database under accession codes GSE211948. The datasets generated and analyzed in the current study are available from the corresponding author, Takashi Tsuji, on request. Source data are provided with this paper.

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

## Acknowledgements

We thank lab members in RIKEN BDR, especially Dr. A. Noma, Ms. H. Sasaki, and Ms. M. Shirasaki, Ms. S. Uchida for technical assistance. We applicate Laboratory for Animal Resources and Genetic Engineering (LARGE), RIKEN Center for Biosystems Dynamics Research to support animal work and provided R26-H2B-EGFP transgenic mice. We also appreciate RIKEN BioResource Research Center (BRC) for providing B6.Cg-Tg(FucciG1)#596Bsi transgenic mice. This work was supported by JSPS KAKENHI (Grant number 21K12678), The Koyanagi-Foundation, The Ohsumi Frontier Science Foundation to Ma.T. We would like to thank OrganTech Inc. for their funding support. We appreciate the generosity of our donors for a RIKEN Solicited donation project, Donations to support basic and applied research for the next generation organ regenerative medicine.

## Author contributions

Ma.T., K.T. and T.T. designed the research plan and Ma. T. coordinate the project. Ma.T., T.I., R.F., Mi.T., K.T., and M.O. performed experiments, analyzed the data, and prepared figures. Ma.T. and T.T. wrote the manuscript.

## Competing interests

The authors declare no competing interests.
