## [Peer Review File · Nature Communications]

Cyclical dermal micro-niche switching governs the morphological infradian rhythm of mouse zigzag hairREVIEWER COMMENTS

Reviewer #1 (Remarks to the Author):

In this manuscript, Takeo et al. have addressed the cyclic formation of bends in zigzag hairs. The authors first thoroughly quantified the position of zigzag hairs in 3D space (and found 2 mirror conformations), the angle of each bend (and found to be approximately 160), and the distance between bends (and found to be approximately 1.5-1.8mm). This constant distance suggests that bend formation occurs at a consistent time interval and thus the authors further went on to evaluate the time of bend formation. The first, second and third bends were estimated to form at 9, 12.3, and 15.8 days post anagen induction.

Using the H2B-EGFP/Fucci596 reporter mouse line, the authors demonstrated two cyclic events that occur during hair shaft formation and between the formation of two adjacent bends:

1. The first event is that a subgroup of matrix cells synchronously enters the G1 phase of the cell cycle and asymmetrically forms a cluster next to the dermal papilla (DP) micro-niche C2. This event cyclically occurs twice after the formation of the first bend and before the formation of the second bend, approximately one day apart.
2. The second event is the cyclic alteration in the position of the C2 and C3 DP micro-niches to partially overlap. This event occurs once, between the formation of the first and second bends immediately before the generation of the second bend.

When the two events coincide (the G1 matrix cluster and the DP micro-niche switch), the G1 cluster clonally moves upwards and relocates next to the forming bend at the upper bulb region. Furthermore, this cluster became thicker and consequently compresses the hair shaft, seemingly affecting the spatial distribution of the medulla. The authors suggest that these cyclical dynamics in the matrix and the DP are involved in bend formation. Using RNA-seq analysis, the authors identified Ptn and Aff3 to be upregulated when these events coincide, and in-situ hybridization revealed that Ptn expression is confined to the G1 cluster and Aff3 expression is limited to the DP C3 micro-niche. Lastly, using hair reconstitution assay combined with the organ germ method, the authors performed loss-of-function analysis to test the role of Ptn and Aff3 in timing of bend formation.

Major concerns:

1. The skin reconstitution assay, as the authors acknowledge, cannot distinguish between the involvement of the genes in morphogenesis and their role in bend formation. The authors claim that by using a chimeric analysis, this intrinsic obstacle is resolved. Such analysis not only does not resolve the problem but also introduces unknown factors that affect the outcome of the experiment and its interpretation. For example, the relative ratio between wild type and mutant cells may affect the proliferation rate of matrix cells and indirectly affect the timing of bend formation. Furthermore, the fact that similar bending defects are also observed in controls at lower frequency suggests that additional factors are involved in these defects which are unrelated to Ptn and Aff3. To really prove that Ptn and Aff3 are involved in the regulation of bend formation, the authors need to conditionally ablate these genes in-vivo using inducible Cre lines specific for the matrix and DP. Complementary to the loss-of-function, the authors should also include gain-of-function analysis. If the author's hypothesis is correct, constitutive expression of Ptn and Aff3 should affect bend formation in a more predictive and informative fashion. In the absence of these data, the manuscript remains descriptive and lacks a mechanistic insight, and thus inadequate for publication in Nature Communications even though the authors discovered an interesting and novel phenomenon.
2. It is not clear how the authors distinguish between zigzag and non-zigzag follicles in Figure 2c. Depending on the strain, non-zigzag follicles at this age represent 40-50% of the follicle population. Does the micro-niche rearrangement occur in zig-zag follicles only? If yes, this will further support

their hypothesis. If not, this will refute their hypothesis. The same argument applies for the formation of the G1 cluster.

Minor concerns:

1. Figure 4d; in-situ for Aff3 does not show any signal in the DP.
2. Introduction second paragraph: No references were included regarding the description of the hair cycle.
3. While the concept of DP micro-niches is probably true, the strict boundaries between the micro-niches have never been established. The authors unnecessarily assume the existence of such boundaries. Their observation is still interesting even if they will restrict their description to reengagement of DP cells without referring to micro-niches.

Reviewer #2 (Remarks to the Author):

The manuscript by Takeo et al investigates how the period bends in mouse zigzag hair form. The authors argue that every three days there is a change in the structure and activity of a subset of dermal papilla cells and hair progenitors that leads to the bend. Using a hair reconstitution assay, the authors implicate Aff3 and Ptn in dermal papilla and hair progenitors, respectively, as causative for this rhythmic bending.

There is substantive work behind this paper and the zigzag hair bends are interesting model to understand infradian rhythms in tissue morphogenesis. Many of the images in the figures and the movies are beautiful.

The main weakness of the paper is that although it describes new morphological features and genes associated with zigzag bending, there is no mechanistic insight into rhythm generation. There are also a number of other significant issues that need to be addressed:

Other major comments:

1. Asymmetric gene expression (such as for Shh) and asymmetric structure of the hair progenitor zone has long been described. But are these features not found in all growing hair follicles rather than being specific for zigzag hair follicles? Hence, it is not clear in many of the figures (Fig. 2-5) how the authors knew they were studying follicles that generate zigzag hairs. And if they had a way of knowing this, it would be important to include other hair follicle types to establish specificity of these phenomena for bending in zigzag follicle.
2. The subtypes of dermal papilla cells defined by Fuchs and colleagues were based on markers. In this manuscript, the authors rely solely on location of the cells. It would significantly strengthen this manuscript to mark the cells.
3. Although the hair reconstitution system generates hair follicles, this is an artificial system. It is therefore important to know whether mouse mutants of Aff3 and Ptn have phenotypes consistent with the authors' model. Both mutants are available and there are multiple publications on Ptn ko mice suggesting that this mutant is widely available. What is the hair phenotype in this mutant. There are also mutations of these genes in the human population. The authors should look into these mutants to the extent they are available.

Other comments:

4. The authors' model is not very clear. Do they propose that the changes in the DP affect the epithelial cells or the other way around? Or, do they propose that there is an unknown signal that affects both compartments simultaneously to start the changes required for bending?
5. Panel 2c is difficult to understand. How do they know that these are all C2 in panel c and how do they know that these are all zigzag follicles? The P12 follicle in the e panel is not multilayered and it seems that the G1 cells are much more widely distributed than implied in the model and in the text.
6. In the PCA in Fig. 4a, it is surprising that based on overall gene expression, P11.5 is more different from P12 than P11 is. Although not impossible, this pattern is not entirely consistent with a rhythm at a three-day time scale. This raises the question of whether enough replicates have been done. What exactly are the replicates here (from different mice, same mouse etc)? The authors need to state how much variance there is in PC1 and PC2.
7. How do the authors account for a half-day fluctuation in Shh expression? Do they propose that Shh has a circadian pattern of gene expression?
8. Discussion: Authors talk about the circadian clock but do not make strong connections between the circadian clock and the morphological clock. Also, throughout the paper, the authors do not present the data as a time series data. So, the part about the circadian clock in the discussion seems to be unnecessarily detailed.
9. The authors refer to "spacing between each medulla." To the best of my knowledge, "medulla" refers to the whole structure and the terms "medulla cells" and "air spaces" have been used to describe the components.
10. The manuscript needs some editing for types, word errors, and style.

Reviewer #3 (Remarks to the Author):

The fur of Murine skin has four distinct hair follicle types, guard, awl, auchene, and zigzag. Zigzag hairs have 3 'kinks/bends' that zig-zag, and are the most abundant hair type in furry animals. The quantity and combination of these hair follicle types can influence how the fur of a mammal appears. Understanding the molecular and cellular mechanisms of hair follicle heterogeneity in murine skin is an ideal model system to investigate stem-cell-niche interactions and tissue rhythms. By investigating how zigzag hairs get their kinks the authors of this manuscript have suggested cyclical growth pattern regulated by Shh, Ptn, and Aff3 in epidermal and mesenchymal cell types. More excitingly, the authors studied hair follicle growth rates in relation to the mechanisms of 'kink' formation in zigzag hairs utilizing 'waxing assays' and the H2B-Fucci transgenic model systems. The basic analysis found that zigzag hair follicles are constructed in the dermis of mice for about 3 days before a burst of progenitor cell division near one side of the hair follicle forces the formation of a kink by slowing down assembly of hair. This process that authors reveal through various experiments is repeated every 3 days to form 2 more kinks of a zigzag hair.

This manuscript is exciting and novel because it approaches understanding stem cell/niche interactions through the formation of distinct structures (kinks in hairs). This has not been investigated for over 10 years. The implications of this study are important for skin regeneration and wound healing studies since controlling epidermal phenotypes are important to consider during organ repair. I have a few suggestions that are required to be addressed before publication to help solidify the molecular results.

Major:

Overall the manuscript was refreshing to read and the data are interesting. However, the major concern is with the figure 5, which utilizes induced genetic mutations in 'bioengineered organs'. I do not find the genetic data convincing because there seems to be a lack of controls and also the small number of hair follicles that were produced from this type of assay. Here are my recommendations for this part of this manuscript.

The authors should increase the number of hair follicles formed in the grafting assay. This could be done by switching to a chamber grafting that was developed by Stuart Yuspa and described in Driskell and Jensen Nature Pro 2011. Or by optimizing the Nakao et al. protocol to be more efficient and increasing the number of hair follicles that are forming.

The authors should present controls of that the Shh, Ptn, and Aff3 are indeed knocked down in the cells that are in the allograft assay.

While not required for publication the authors could attempt to get cells from a knockout mouse model. The Aff3flx/flx mice have been published and would make for better model system to knockout these genes.

Minor:

The authors should scan the manuscript for typos and misspelled words. There are many mistakes throughout the manuscript and in the figures themselves. (For example in Figure 5c ermal cells?)

Response to reviewers

REVIEWER COMMENTS

Reviewer #1 (Remarks to the Author):

In this manuscript, Takeo et al. have addressed the cyclic formation of bends in zigzag hairs. The authors first thoroughly quantified the position of zigzag hairs in 3D space (and found 2 mirror conformations), the angle of each bend (and found to be approximately 160), and the distance between bends (and found to be approximately 1.5-1.8mm). This constant distance suggests that bend formation occurs at a consistent time interval and thus the authors further went on to evaluate the time of bend formation. The first, second and third bends were estimated to form at 9, 12.3, and 15.8 days post anagen induction.

Using the H2B-EGFP/Fucci596 reporter mouse line, the authors demonstrated two cyclic events that occur during hair shaft formation and between the formation of two adjacent bends:

1. The first event is that a subgroup of matrix cells synchronously enters the G1 phase of the cell cycle and asymmetrically forms a cluster next to the dermal papilla (DP) micro-niche C2. This event cyclically occurs twice after the formation of the first bend and before the formation of the second bend, approximately one day apart.
2. The second event is the cyclic alteration in the position of the C2 and C3 DP micro-niches to partially overlap. This event occurs once, between the formation of the first and second bends immediately before the generation of the second bend.

When the two events coincide (the G1 matrix cluster and the DP micro-niche switch), the G1 cluster clonally moves upwards and relocates next to the forming bend at the upper bulb region. Furthermore, this cluster became thicker and consequently compresses the hair shaft, seemingly affecting the spatial distribution of the medulla. The authors suggest that these cyclical dynamics in the matrix and the DP are involved in bend formation. Using RNA-seq analysis, the authors identified Ptn and Aff3 to be upregulated when this events coincide, and *in-situ* hybridization revealed that Ptn expression is confined to the G1 cluster and Aff3 expression is limited to the DP C3 micro-niche. Lastly, using hair reconstitution assay combined with the organ germ method, the authors performed loss-of-function analysis to test the role of Ptn and Aff3 in timing of bend formation.

Our Response:

We are grateful for your careful reading of our manuscript. According to your comments and suggestion, we have performed a gain-of-function assay and analysis of DP cell rearrangement in types of hair follicles other than zigzag hair, improved the figure for *in situ* hybridization, and added a reference for the hair cycle.

Major concerns:

1. The skin reconstitution assay, as the authors acknowledge, cannot distinguish between the involvement of the genes in morphogenesis and their role in bend formation. The authors claim that by using a chimeric analysis, this intrinsic obstacle is resolved. Such analysis not only does not resolve the problem but also introduces unknown factors that affect the outcome of the experiment and its interpretation.

For example, the relative ratio between wild type and mutant cells may affect the proliferation rate of matrix cells and indirectly affect the timing of bend formation.

Furthermore, the fact that similar bending defects are also observed in controls at lower frequency suggests that additional factors are involved in these defects which are unrelated to Ptn and Aff3.

To really prove that Ptn and Aff3 are involved in the regulation of bend formation, the authors need to conditionally ablate these genes in-vivo using inducible Cre lines specific for the matrix and DP.

Complementary to the loss-of-function, the authors should also include gain-of-function analysis. If the author's hypothesis is correct, constitutive expression of Ptn and Aff3 should affect bend formation in a more predictive and informative fashion. In the absence of these data, the manuscript remains descriptive and lacks a mechanistic insight, and thus inadequate for publication in Nature Communications even though the authors discovered an interesting and novel phenomenon.

Our Response:

As the reviewer noted, we could reach clearer and more reliable conclusions by using a mouse model that can specifically target the matrix and DP. However, currently, no such mouse models are available. Several Cre lines, such as Sox2-CreER and Pdgfra-CreER, can target DPs of only guard, aul, and auchene hair follicles but not zigzag hair

follicles. Cre lines that target epithelial cells, such as K14-CreER and K5-CreER, affect all basal epithelial cells, similar to our method using CRISPR and hair reconstitution assays. Therefore, we decided to use an *in vivo* hair reconstitution assay rather than *in vivo* mouse models to narrow down the target cells.

In KO experiments, we observed a defect in the angle of bend in the control group, while a defect in the timing of bend formation occurred in the *Ptn* and *Aff3* KO groups, demonstrating that CRISPR itself does not affect the timing of bend formation (Fig. 5f and g). Although *Aff3* KO targets dermal cells but not hair matrix cells, which give rise to IRS involved in bend formation, we still observed a defect in the timing of bend formation (Fig. 5f and g).

In addition, according to the reviewer's suggestion, we performed a hair reconstitution assay using epidermal or dermal cells that constitutively expressed the *Shh*, *Ptn*, or *Aff3* gene. We observed no hair follicle formation when *Shh* was overexpressed in epidermal cells, consistent with a previous study (Ellis, T. *et al.* 2003. *Dev. Bio.* 263, 203-215, doi:10.1016/s0012-1606(03)00394-4). In *Ptn* overexpression, hair shafts show 2 or fewer bends, a thin diameter, and sparse distribution of the medulla cells, which might be due to the uniform expression of *Ptn* in the hair matrix leading to the uniform thickening of inner root sheath. In contrast, overexpression of *Aff3* in dermal cells resulted in an increased number of bends, which might be due to the contact of G1-HM with *Aff3*-overexpressing DP more frequently (Fig. 6).

Taken together, these results at least indicate that the defect in bend formation seen in the control group results from artifacts or unknown factors, while the defects in the timing of bend formation seen in the *Ptn* and *Aff3* KO groups as well as the overexpression group are due to the disturbance of *Ptn* and *Aff3* gene function rather than to differences in proliferation rate between wild-type matrix cells. In other words, these data clearly suggested that at least *Ptn* expression in epithelial cells, including the matrix, and *Aff3* expression in dermal cells, including DP cells, are involved in determining the timing of bend formation.

Although the mechanism that controls DP rearrangement and the expression of *Ptn* and *Aff3* has to be further investigated using a suitable mouse model in the future, our data demonstrate the partial cellular and molecular mechanism of bend formation. We appreciate the reviewer's very useful comments, which made our data and claim stronger and clearer. We believe our manuscript is now suitable for publication in *Nature Communications*.

2. It is not clear how the authors distinguish between zigzag and non-zigzag follicles in Figure 2c. Depending on the strain, non-zigzag follicles at this age represent 40-50% of the follicle population. Does the micro-niche rearrangement occur in zig-zag follicles only? If yes, this will further support their hypothesis. If not, this will refute their hypothesis. The same argument applies for the formation of the G1 cluster.

Our Response:

In Fig. 2c, we distinguish zigzag and non-zigzag hair follicles by the number of columns of medulla cells and the presence of the bend using a higher resolution image, as seen in Fig. S3d. In general, zigzag hair has only one column of medulla cells, while other hair types have multiple columns. Moreover, guard and awl hair have no bend, and auchene hair has only one bend. In this study, we analyzed the 2nd bend formation of zigzag hair since it can be definitely recognized as zigzag hair by the presence of the first bend point. To show this point, we added photographs of typical hair shafts of guard, awl, and auchene hair as Fig. S1 and explanations in the main text.

Based on the above criteria, we analyzed the formation of G1-HM and micro-niche rearrangement in non-zigzag hair follicles. We performed quantitative analysis of the number of G1-HM facing the dermal papilla only on the awl hair because guard, awl, and auchene hair follicles represent 1-3%, 30%, and 0.1%, respectively, of the total number of hair, and we could not obtain a statistically sufficient number of hair follicles for guard and auchene hair follicles (Fig. S3c). We found that the number of G1-HM facing the dermal papilla was always less than 6 cells in the awl hair follicle, while it increased by more than 6 cells at P12.0 dpd in the zigzag hair follicle. Moreover, detailed spatiotemporal analysis of micro-niche cells by confocal microscopy revealed no significant rearrangement of DP micro-niche in awl hair follicles (Fig. S4). These results indicate that the formation of the G1-HM cluster and rearrangement of the DP micro-niche is a phenomenon specific to the zigzag hair follicle.

Minor concerns:

1. Figure 4d; *in-situ* for Aff3 does not show any signal in the DP.

Our Response:

To make the *in situ* signal clear, we added a cross-sectional picture of the region where *Aff3* is expressed.

2. Introduction second paragraph: No references were included regarding the description of the hair cycle.

Our Response:

According to the reviewer's comment, we added the following reference to the Introduction.

Muller-Rover *et al.*, *J Invest Dermatol.* 2001 Jul;117(1):3-15.

3. While the concept of DP micro-niches is probably true, the strict boundaries between the micro-niches have never been established. The authors unnecessarily assume the existence of such boundaries. Their observation is still interesting even if they will restrict their description to reengagement of DP cells without referring to micro-niches.

Our Response:

We appreciate the kind comment from the reviewer. According to the comment from another reviewer, we analyzed markers for DP subtypes C1, C2, and C3 by immunohistochemistry and *in situ* hybridization according to a previous report (Yang, H. *et al.*, 2017. *Cell* 169, 483-496 e413, doi:10.1016/j.cell.2017.03.038). In contrast to the previous report, markers for DP subtypes 1 and 2 were detectable throughout the dermal papilla, and no specific pattern was observed. In contrast, the marker for DP subtype C3, which is a key micro-niche cluster in this study, is expressed in a subset of DP cells, the boundaries of which coincide with the boundaries of DP cell clusters (Fig. 3e). We also observed rearrangement of the DP cluster expressing a marker for DP subtype C3 at P12.0 dpd immediately before bend formation, clearly demonstrating the concept of micro-niche switching (Fig. 3e).

Reviewer #2 (Remarks to the Author):

The manuscript by Takeo et al investigates how the period bends in mouse zigzag hair form. The authors argue that every three days there is a change in the structure and activity of a subset of dermal papilla cells and hair progenitors that leads to the bend. Using a hair reconstitution assay, the authors implicate *Aff3* and *Ptn* in dermal papilla and hair progenitors, respectively, as causative for this rhythmic bending.

There is substantive work behind this paper and the zigzag hair bends are interesting model to understand infradian rhythms in tissue morphogenesis. Many of the images in the figures and the movies are beautiful.

The main weakness of the paper is that although it describes new morphological features and genes associated with zigzag bending, there is no mechanistic insight into rhythm generation.

Our Response:

We appreciate your careful reading of our manuscript and acknowledge our research as an interesting phenomenon related to the generation of infradian rhythm in tissue morphogenesis.

As the reviewer pointed out, our current study cannot elucidate the detailed mechanism of the generation of the infradian rhythm. However, we demonstrated that the G1-HM cluster is asymmetrically formed in the bulb region once a day and that DP micro-niche rearrangement occurs once in approximately 3 days, immediately before bend formation (Figs. 2 and 3). We also showed that G1-HM cluster formation resulted in thickening of the inner root sheath of the squash hair shaft (Fig. 2). We demonstrated that gene knockout of *Ptn* and *Aff3* leads to less bend formation and large diversity in the distance between the hair tip and each bend (Fig. 5). Moreover, according to the reviewers' comment, we performed a gain-of-function assay by forced activation of *Shh* and *Ptn* in epidermal cells and *Aff3* in dermal cells by using a lentivirus system. We observed no hair follicle formation when *Shh* was overexpressed in epidermal cells, consistent with a previous study (Ellis, T. *et al.* 2003. *Dev. Bio.l* 263, 203-215, doi:10.1016/s0012-1606(03)00394-4). In *Ptn* overexpression, hair shafts show 2 or fewer bends, a thin diameter, and sparse distribution of the medulla cells, which might be due to the uniform

expression of *Ptn* in the hair matrix leading to the uniform thickening of inner root sheath. In contrast, overexpression of *Aff3* in dermal cells resulted in an increased number of bends. These results suggest that the contact of the *Aff3*-expressing DP micro-niche with *Ptn*-expressing G1-HM changes the cell dynamics of IRS and forms bend points. Although the mechanism that controls DP rearrangement and the expression of *Ptn* and *Aff3* has to be further investigated in the future, we believe that our data demonstrate the partial cellular and molecular mechanisms that generate the rhythm of bend formation and are suitable for publication in *Nature Communications*.

There are also a number of other significant issues that need to be addressed:

Other major comments:

1. Asymmetric gene expression (such as for *Shh*) and asymmetric structure of the hair progenitor zone has long been described. But are these features not found in all growing hair follicles rather than being specific for zigzag hair follicles? Hence, it is not clear in many of the figures (Fig. 2-5) how the authors knew they were studying follicles that generate zigzag hairs. And if they had a way of knowing this, it would be important to include other hair follicle types to establish specificity of these phenomena for bending in zigzag follicle.

Our Response:

We appreciate the reviewer raising this point. In Figs. 2-5, we distinguish zigzag and non-zigzag hair follicles by the number of columns of medulla cells and the presence of the bend. In general, zigzag hair has only one column of medulla cells, while other hair types have multiple columns. Moreover, guard and awl hair have no bend, and auchene hair has only one bend. In this study, we analyzed the 2nd bend formation of zigzag hair since it can be definitely recognized as zigzag hair by the presence of the first bend point. To show this point, we added photographs of typical hair shafts of guard, awl, and auchene hair as Fig. S1 and explain in the main text.

To examine whether asymmetric G1-HM is specific for zigzag hair follicles, we analyzed the formation of G1-HM in non-zigzag hair follicles. We performed quantitative analysis of the number of G1-HM facing the dermal papilla only on the awl hair because

guard, awl, and auchene hair follicles represent 1-3%, 30%, and 0.1%, respectively, of the total number of hair, and we could not obtain a statistically sufficient number of hair follicles for guard and auchene hair follicles (Fig. S3c). We found that the number of G1-HM facing the dermal papilla was always less than 6 cells in the awl hair follicle, while it increased by more than 6 cells at P12.0 dpd in the zigzag hair follicle. These results suggest that the formation of G1-HM clusters is a phenomenon specific to zigzag hair follicles.

2. The subtypes of dermal papilla cells defined by Fuchs and colleagues were based on markers. In this manuscript, the authors rely solely on location of the cells. It would significantly strengthen this manuscript to mark the cells.

Our Response:

We appreciate the suggestion from the reviewer to analyze DP micro-niche subtype based on markers.

We analyzed markers for DP subtypes C1, C2, and C3 by immunohistochemistry and *in situ* hybridization according to a previous report (Yang, H. *et al.*, 2017. *Cell* **169**, 483-496 e413, doi:10.1016/j.cell.2017.03.038). In contrast to the previous report, markers for DP subtypes 1 and 2 were detectable throughout the dermal papilla, and no specific pattern was observed. In contrast, the marker for DP subtype C3, which is a key micro-niche cluster in this study, is expressed in a subset of DP cells, the boundaries of which coincide with the boundaries of DP cell clusters (Fig. 3e). We also observed rearrangement of the DP cluster expressing a marker for DP subtype C3 at P12.0 dpd immediately before bend formation, clearly demonstrating the concept of micro-niche switching (Fig. 3e).

3. Although the hair reconstitution system generates hair follicles, this is an artificial system. It is therefore important to know whether mouse mutants of Aff3 and Ptn have phenotypes consistent with the authors' model. Both mutants are available and there are multiple publications on Ptn ko mice suggesting that this mutant is widely available. What is the hair phenotype in this mutant. There are also mutations of these genes in the human population. The authors should look into these mutants to the extent they are available.

Our Response:

Based on the literature, mutant mice for *Ptn* and *Aff3* show phenotypes not only in hair follicles but also in multiple organs, which makes it difficult to properly assess the effect of mutation on bend formation. Therefore, in this study, we decided to use a hair reconstitution assay in combination with CRISPR-mediated gene knockout for functional analysis.

Although the hair reconstitution assay is an artificial system, regenerated hair follicles are equivalent to natural hair follicles (Toyoshima *et al.*, 2012. *Nat Commun.* 3:784). Moreover, as described above, in KO and gain-of-function assays, we observed defects in the timing of bend formation in the *Ptn* and *Aff3* groups but not in the control group. These results strongly suggest that the hair reconstitution assay system works at least to examine the role of *Ptn* and *Aff3* in the timing of bend formation, and the obtained result is reliable.

We appreciate the interesting suggestion of the reviewer about *Ptn* and *Aff3* mutations in humans. We are interested in the function of these genes in human hair formation and would like to look into them in the future.

Other comments:

4. The authors' model is not very clear. Do they propose that the changes in the DP affect the epithelial cells or the other way around? Or, do they propose that there is an unknown signal that affects both compartments simultaneously to start the changes required for bending?

Our Response:

As mentioned above, based on the loss-of-function and gain-of-function assays, we proposed that the contact of the *Aff3*-expressing DP micro-niche with *Ptn*-expressing Gi-HM changes the cell dynamics of IRS and forms bend points. We think there are unknown signals that control DP rearrangement and the expression of *Ptn* and *Aff3*, and we have to identify this signal in future studies.

5. Panel 2c is difficult to understand. How do they know that these are all C2 in panel c and how to they know that these are all zigzag follicles? The P12 follicle in the e panel is

not multilayered and it seems that the G1 cells are much more widely distributed than implied in the model and in the text.

Our Response:

Although it is difficult to know which micro-niche of dermal papillae we are analyzing, we can at least distinguish multilayered dermal papilla from monolayered dermal papilla by using a higher resolution image, as shown in Fig. S3d. We revised the main text and figure labels to not mention dermal papilla as C2.

To distinguish zigzag hair follicles and non-zigzag hair follicles, as mentioned above, we evaluated the number of columns of medulla cells and the presence of the bend. In this study, we analyzed the 2nd bend formation of zigzag hair since it can be definitely recognized as zigzag hair by the presence of the first bend point.

For Panel 2e, why the dermal papilla does not look multilayered is because we are looking at different positions than the multilayered position. The purpose of this panel is to show that G1-HM is formed asymmetrically across the dermal papilla.

As the reviewer mentioned, G1 cells are widely distributed in the hair follicle. In the proposed model and the text, we only described G1 cells facing the dermal papilla; therefore, the distribution of G1 cells in low data and the distribution in the model and the text are different.

6. In the PCA in Fig. 4a, it is surprising that based on overall gene expression, P11.5 is more different from P12 than P11 is. Although not impossible, this pattern is not entirely consistent with a rhythm at a three-day time scale. This raises the question of whether enough replicates have been done. What exactly are the replicates here (from different mice, same mouse etc)? The authors need to state how much variance there is in PC1 and PC2.

Our Response:

In this study, we performed RNA-Seq analysis on two replicates for each time point, and each replicate contained at least 30 hair bulbs isolated from a single mouse. The variances of PC1 and PC2 are 99.89% and 0.07%, respectively.

In RNA-Seq analysis, we analyzed the gene profile for 1.5 days from P11 to P12. Therefore, we could not see the pattern with a 3-day interval. During bend formation for the entire 3 days, we think the gene expression profile oscillates once a day concomitant

with G1-HM formation. Consistent with this hypothesis, samples isolated from P11 and P12 had close locations in the PCA. Moreover, more genes should fluctuate more at P12 for bend formation; therefore, P11.5 differs more from P12 than P11.

7. How do the authors account for a half-day fluctuation in Shh expression? Do they propose that Shh has a circadian pattern of gene expression?

Our Response:

Previous studies have demonstrated that Shh is a target of circadian genes and that circadian genes oscillate in the hair matrix (Sukumaran, S., *et al.*, 2010. *Adv Drug Deliv Rev* 62, 904-917. Plikus *et al.*, 2013. *Proc Natl Acad Sci USA*. 110(23):E2106-15). Based on these facts, we think Shh is expressed in a circadian manner. We added this point to the main text.

8. Discussion: Authors talk about the circadian clock but do not make strong connections between the circadian clock and the morphological clock. Also, throughout the paper, the authors do not present the data as a time series data. So, the part about the circadian clock in the discussion seems to be unnecessarily detailed.

Our Response:

According to the reviewer's comment, we rewrote the discussion about circadian rhythm more simply.

9. The authors refer to "spacing between each medulla." To the best of my knowledge, "medulla" refers to the whole structure and the terms "medulla cells" and "air spaces" have been used to describe the components.

Our Response:

According to the reviewer's comment, we used "medulla cells" throughout the manuscript.

10. The manuscript needs some editing for typos, word errors, and style.

Our Response:

We have corrected the typo, word errors, and style.

Reviewer #3 (Remarks to the Author):

The fur of Murine skin has four distinct hair follicle types, guard, awl, auchene, and zigzag. Zigzag hairs have 3 'kinks/bends' that zig-zag, and are the most abundant hair type in furry animals. The quantity and combination of these hair follicle types can influence how the fur of a mammal appears. Understanding the molecular and cellular mechanisms of hair follicle heterogeneity in murine skin is an ideal model system to investigate stem-cell-niche interactions and tissue rhythms. By investigating how zigzag hairs get their kinks the authors of this manuscript have suggested cyclical growth pattern regulated by Shh, Ptn, and Aff3 in epidermal and mesenchymal cell types. More excitingly, the authors studied hair follicle growth rates in relation to the mechanisms of 'kink' formation in zigzag hairs utilizing 'waxing assays' and the H2B-Fucci transgenic model systems. The basic analysis found that zigzag hair follicles are constructed in the dermis of mice for about 3 days before a burst of progenitor cell division near one side of the hair follicle forces the formation of a kink by slowing down assembly of hair. This process that authors reveal through various experiments is repeated every 3 days to form 2 more kinks of a zigzag hair.

This manuscript is exciting and novel because it approaches understanding stem cell/niche interactions through the formation of distinct structures (kinks in hairs). This has not been investigated for over 10 years. The implications of this study are important for skin regeneration and wound healing studies since controlling epidermal phenotypes are important to consider during organ repair. I have a few suggestions that are required to be addressed before publication to help solidify the molecular results.

Our Response:

We are grateful that the reviewer noted the importance and novelty of our study, especially regarding the stem cell-niche interaction during hair morphogenesis. According to the reviewer's comments, we added new data to confirm the presence of the CRISPR-mediated gene KO allele in the regenerated hair follicle and increase the number of hair follicles we analyzed in Fig. 5.

Major:

Overall the manuscript was refreshing to read and the data are interesting. However, the major concern is with the figure 5, which utilizes induced genetic mutations in

'bioengineered organs'. I do not find the genetic data convincing because there seems to be a lack of controls and also the small number of hair follicles that were produced from this type of assay. Here are my recommendations for this part of this manuscript.

The authors should increase the number of hair follicles formed in the grafting assay. This could be done by switching to a chamber grafting that was developed by Stuart Yuspa and described in Driskell and Jensen Nature Pro 2011. Or by optimizing the Nakao et al. protocol to be more efficient and increasing the number of hair follicles that are forming.

Our Response:

We appreciate the comment on the method of the grafting assay. In this study, as the reviewer suggested in his or her comment, we performed hair reconstitution according to Nakao's protocol. The advantage of this method is that hair follicles can be regenerated with a small number of cells, and the number of regenerated hair follicles can be controlled (Toyoshima *et al.*, 2012. *Nat Commun.* 3:784). We can increase the number of regenerated hairs if we want to, but to isolate and analyze hair follicles one by one, we deliberately regenerated a small number of hairs from a single shot of hair reconstitution. To increase the number of hair follicles analyzed, we repeated the same grafting experiment and increased the number of hair follicles by 3 times.

For genetic evaluation of CRISPR-mediated gene KO, we performed Cycleave PCR, which can detect single base differences, and subsequent DNA sequencing. We successfully detected a single base deletion in *Shh*, *Ptn*, and *Aff3* KO hair follicles, suggesting that these genes are knocked out at least in the subset of hair follicle cells (Fig. S5).

The authors should present controls of that the *Shh*, *Ptn*, and *Aff3* are indeed knocked down in the cells that are in the allograft assay.

Our Response:

As shown in Fig. 4c, the mRNA levels of *Shh*, *Ptn*, and *Aff3* varied even at the same time point. Moreover, as shown in Fig. 5b, guide RNAs were not incorporated into all epidermal or dermal cells in the CRISPR experiment. For these reasons, it is not possible to accurately determine whether the gene is knocked out by simply comparing the expression levels. Therefore, we decided to detect the knockout allele instead of mRNA.

By Cyclave PCR and subsequent DNA sequencing of regenerated hair follicles, we detected single base deletions in the *Shh*, *Ptn*, and *Aff3* loci, which disturb normal gene expression. Based on this observation, we think these genes are knocked out at least in the subset of hair follicle cells. We have added these data as Fig. S5.

While not required for publication the authors could attempt to get cells from a knockout mouse model. The *Aff3*flx/flx mice have been published and would make for better model system to knockout these genes.

Our Response:

As the reviewer suggested, we could reach clearer and more reliable conclusions by using a mouse model that can specifically target the matrix and DP. However, although *Aff3* flx/flx mice have been published, Cre driver mice, which specifically target the hair matrix or DP of zigzag hair follicles, are not available. Several Cre lines, such as *Sox2*-CreER and *Pdgfra*-CreER, can target DPs of only guard, aul, and auchene hair follicles but not zigzag hair follicles. Cre lines that target epithelial cells, such as *K14*-CreER and *K5*-CreER, affect all basal epithelial cells. Therefore, we decided to use an *in vivo* hair reconstitution assay rather than *in vivo* mouse models to narrow down the target cells. In the hair reconstitution assay, we observed a defect in the angle of bend in the control group, while a defect in the timing of bend formation occurred in the *Ptn* and *Aff3* KO groups. These results suggest that at least *Ptn* expression in epithelial cells, including the matrix, and *Aff3* expression in dermal cells, including DP cells, are involved in determining the timing of bend formation.

To further analyze the function of *Ptn* and *Aff3* in detail, we would like to generate and use a suitable mouse model in future experiments.

Minor:

The authors should scan the manuscript for typos and misspelled words. There are many mistakes throughout the manuscript and in the figures themselves. (For example, in Figure 5c ermal cells?)

Our Response:

We have scanned the entire manuscript and identified typos and misspelled words.

REVIEWERS' COMMENTS

Reviewer #1 (Remarks to the Author):

While the authors made the effort to improve the manuscript by including overexpression analysis, the major concern regarding the hair reconstitution assay remained unanswered and therefore the study lacks a mechanistic insight. The hair reconstitution assay is completely artificial and does not reflect the physiological process of zigzag hair formation.

In their rebuttal, the authors claim: "Cre lines that target epithelial cells, such as K14-CreER and K5-CreER, affect all basal epithelial cells, similar to our method using CRISPR and hair reconstitution assays. Therefore, we decided to use an in vivo hair reconstitution assay rather than in vivo mouse models to narrow down the target cells". I respectfully disagree; while both methods target the same compartment, their interpretations are fundamentally different. However, since a floxed line for Ptn is not available and since Ptn expression is restricted to a subpopulation of matrix cells, the authors could at least analyze the available knockout mice (L. E. A. Amet et al. 2001, Molecular and Cellular Neuroscience). Such analysis will allow the authors to include depilation and systematic follow-up as they conducted for the wild type mice. Similarly, the authors also claim: "As the reviewer noted, we could reach clearer and more reliable conclusions by using a mouse model that can specifically target the matrix and DP. However, currently, no such mouse models are available. Several Cre lines, such as Sox2-CreER and Pdgfra-CreER, can target DPs of only guard, aul, and auchene hair follicles but not zigzag hair follicles". Since a knockout mouse line for Aff3 is available (Shin-ichi Tsukumo et al. 2022, Sci.Adv) and Aff3 expression is restricted to only a subgroup of DP cells, here too the authors could assess the hair phenotype of these knockout mice.

The authors also claim in the manuscript; "Furthermore, although it is possible that the relative ratio between wild-type and mutant cells may affect the proliferation rate of matrix cells and indirectly affect the timing of bend formation, similar phenotypes were observed when targeting either keratinocytes for Shh and Ptn or fibroblasts for Aff3. These results suggest that the defect in the timing of bend formation seen in the Shh, Ptn and Aff3 KO groups is not just an artifact and that Shh, Ptn and Aff3 have a role in determining the timing of bend formation". I respectfully disagree since similar phenotypes do not refute the possibility that all three are involved in regulating matrix cell proliferation that somehow interferes with the process of bending. In summary, the skin reconstitution assay with its chimeric configuration is not adequate to distinguish between alternative hypotheses. In addition, the overexpression analysis seems incomplete. The authors did not provide images of the bulbs to illustrate the effect of the overexpression on G1-HM clustering and DP micro-niche rearrangement.

Reviewer #2 (Remarks to the Author):

The authors have adequately addressed my comments.

Reviewer #3 (Remarks to the Author):

The authors have addressed my concerns and have improved the manuscript.

Response to reviewers

REVIEWER COMMENTS

Reviewer #1 (Remarks to the Author):

While the authors made the effort to improve the manuscript by including overexpression analysis, the major concern regarding the hair reconstitution assay remained unanswered and therefore the study lacks a mechanistic insight. The hair reconstitution assay is completely artificial and does not reflect the physiological process of zigzag hair formation.

Our Responses:

We appreciate reviewer for kindly reviewing our manuscript and giving a comment, which enable us to reach clearer conclusion. Although the hair reconstitution assay is an artificial system, regenerated hair follicles are equivalent to natural hair follicles (Toyoshima *et al.*, 2012. *Nat Commun.* 3:784). Moreover, in KO and gain-of-function assays, we observed defects in the timing of bend formation in the *Ptn* and *Aff3* groups but not in the control group. These results strongly suggest that the hair reconstitution assay system works at least to examine the role of *Ptn* and *Aff3* in the timing of bend formation, and the obtained result is reliable, although there are several limitations.

In their rebuttal, the authors claim: “Cre lines that target epithelial cells, such as K14-CreER and K5-CreER, affect all basal epithelial cells, similar to our method using CRISPR and hair reconstitution assays. Therefore, we decided to use an in vivo hair reconstitution assay rather than in vivo mouse models to narrow down the target cells”. I respectfully disagree; while both methods target the same compartment, their interpretations are fundamentally different. However, since a floxed line for *Ptn* is not available and since *Ptn* expression is restricted to a subpopulation of matrix cells, the authors could at least analyze the available knockout mice (L. E. A. Amet et al. 2001, *Molecular and Cellular Neuroscience*). Such analysis will allow the authors to include depilation and systematic follow-up as they conducted for the wild type mice. Similarly, the authors also claim: “As the reviewer noted, we could reach clearer and more reliable conclusions by using a mouse model that can specifically target the matrix and DP. However, currently, no such mouse models are available. Several Cre lines, such as Sox2-

CreER and Pdgfra-CreER, can target DPs of only guard, aul, and auchene hair follicles but not zigzag hair follicles”. Since a knockout mouse line for Aff3 is available (Shin-ichi Tsukumo et al. 2022, Sci.Adv) and Aff3 expression is restricted to only a subgroup of DP cells, here too the authors could assess the hair phenotype of these knockout mice.

Our Responses:

We appreciate for the reviewer’s helpful suggestion. As reviewer suggested mutant mice of *Ptn* and *Aff3* are potentially useful model to assess the role of these cells on the bend formation. However, based on the literature, mutant mice for *Ptn* and *Aff3* show phenotypes in multiple organs, which may make it difficult to properly assess the effect of mutation on hair phenotype. To reliably assess the function of *Ptn* and *Aff3* on the bend formation *in vivo*, we have to use the Cre mouse lines to specifically target the G1-HM or micro-niche C3, which are not available for now. We appreciate the useful suggestion of the reviewer about *Ptn* and *Aff3* mutants. In the future study, we would like to further analyze the role of *Ptn* and *Aff3* by generating Cre mouse lines, which can specifically target the G1-HM or micro-niche C3

The authors also claim in the manuscript; “Furthermore, although it is possible that the relative ratio between wild-type and mutant cells may affect the proliferation rate of matrix cells and indirectly affect the timing of bend formation, similar phenotypes were observed when targeting either keratinocytes for Shh and Ptn or fibroblasts for Aff3. These results suggest that the defect in the timing of bend formation seen in the Shh, Ptn and Aff3 KO groups is not just an artifact and that Shh, Ptn and Aff3 have a role in determining the timing of bend formation”. I respectfully disagree since similar phenotypes do not refute the possibility that all three are involved in regulating matrix cell proliferation that somehow interferes with the process of bending. In summary, the skin reconstitution assay with its chimeric configuration is not adequate to distinguish between alternative hypotheses. In addition, the overexpression analysis seems incomplete. The authors did not provide images of the bulbs to illustrate the effect of the overexpression on G1-HM clustering and DP micro-niche rearrangement.

Our Responses:

We appreciate reviewer for pointing this out. As reviewer pointed out, our current data cannot completely rule out the involvement of the difference of the cell proliferation rate between WT and targeted cells in bend formation. Moreover, editor comment on this

point and asked us to acknowledge the possibility that differences in matrix proliferation as a potential contributing factor in the Ptn/Aff3 phenotypes. Therefore, we revised the main text, which reviewer mentioned above, as follows, “These results suggest that Shh, Ptn and Aff3 have a role in determining the timing of bend formation potentially through affecting the rate of matrix cell proliferation between wildtype and KO cells because of chimeric KO assay (page 9, lines 30-32).”

As reviewer suggested, the data about G1-HM clustering and DP micro-niche rearrangement in overexpression experiment will be informative and will help our understanding about the mechanism underlying bend formation. However, in current method, we cannot target all cells and therefore cannot distinguish overexpressed hair follicle from others. Thus, in this assay, we analyzed hair phenotypes and the appearance rate of affected hair shaft. In future study, we would like to use mouse model to target desired cell populations and analyze G1-HM clustering and DP micro-niche rearrangement to further understand the mechanism of bend formation.

We appreciate again reviewer’s time for reviewing our manuscript and give us a useful comment. We hope our response to be satisfactory to reviewers.

Reviewer #2 (Remarks to the Author):

The authors have adequately addressed my comments.

Our Response:

We are very grateful to the reviewer for spending his/her effort and time to careful review of our manuscript.

Reviewer #3 (Remarks to the Author):

The authors have addressed my concerns and have improved the manuscript.

Our Response:

We are very grateful to the reviewer for spending his/her effort and time to careful review of our manuscript.